# Q&C: When Quantization Meets Cache in Efficient Generation

**Xin Ding[1]  Xin Li[1]***  **Haotong Qin[2]  Zhibo Chen[13]***
[1]University of Science and Technology of China   [2] ETH Zürich
[3] Zhongguancun Academy

## Abstract

Quantization and cache mechanisms are typically applied individually in efficient generation tasks, each showing notable potential for acceleration. However, their joint effect on efficiency remains under-explored. Through both empirical investigation and theoretical analysis, we find that that combining quantization with caching is non-trivial, as it introduces two major challenges that severely degrade performance: (i) the sample efficacy of calibration datasets in post-training quantization (PTQ) is significantly eliminated by cache operation; (ii) the joint use of the two mechanisms exacerbates exposure bias in the sampling distribution, leading to amplified error accumulation during generation. In this work, we take advantage of these two acceleration mechanisms and propose a hybrid acceleration method by tackling the above challenges, aiming to further improve the efficiency of tasks while maintaining excellent generation capability. Concretely, a temporal-aware parallel clustering (TAP) is designed to dynamically improve the sample selection efficacy for the calibration within PTQ for different diffusion steps. A variance compensation (VC) strategy is derived to correct the sampling distribution. It mitigates exposure bias through an adaptive correction factor generation. Extensive experiments demonstrate that our method is broadly applicable to diverse generation tasks, achieving up to $12.7\times$ acceleration while preserving competitive generation quality.

## 1 Introduction

The rapid progress of generative models has driven significant breakthroughs across a wide range of tasks, such as image generation (Croitoru et al., 2023; Yang et al., 2023; Peebles & Xie, 2023), video generation (Liu et al., 2024b; Wang et al., 2025). However, their widespread adoption is hindered by the immense computational complexity and large parameter counts. For instance, generating a $512\times512$ resolution image using DiTs can take more than 20 seconds and 105 Gflops on an NVIDIA RTX A6000 GPU (Wu et al., 2024). This substantial requirement makes them unacceptable or impractical for real-time applications, especially as model sizes and resolutions continue to increase (Liu & Zhang, 2024; Zhao et al., 2024a).

Quantization (Nagel et al., 2021; 2020; Liu et al., 2023) and cache (Ma et al., 2024b; Wimbauer et al., 2024; So et al., 2023) have been independently explored as two effective acceleration mechanisms to alleviate the computational burden of generative models (Wu et al., 2024; Ma et al., 2024a; Selvaraju et al., 2024; Chen et al., 2024b). Quantization accelerates inference by converting weights and activations into lower-bit representations, thereby reducing both runtime and memory usage. Among different paradigms, post-training quantization (PTQ) (He et al., 2023) has received particular attention, as it eliminates quantization errors using only a small calibration dataset, making it more resource-efficient compared with quantization-aware training (QAT) (Lu et al., 2024). In contrast, the cache mechanism accelerates generative models by exploiting the reusability of historical features in the diffusion process, thereby avoiding redundant computations during inference. Common strategies leverage the repetitive nature of denoising steps, storing and reusing intermediate representations—such as those from attention and MLP layers—across different timesteps.

Despite the effectiveness of both quantization and cache mechanisms, it remains under-explored "whether integrating these two mechanisms can further boost the efficiency of generative models?"

---

*Corresponding author: xin.li@ustc.edu.cn, chenzhibo@ustc.edu.cn

However, when quantization meets cache, a notable decline in generation quality is often observed, even though impressive acceleration gains can be achieved. To investigate this phenomenon, we conduct an in-depth analysis of the sampling process and identify two crucial factors contributing to the performance degradation.

Firstly, as shown in Figure 1, we find that cache operation dramatically increases sample similarity in the calibration dataset used for PTQ, leading to a marked reduction in sample efficacy. This effect becomes increasingly severe with the number of generative steps, compromising the effectiveness of PTQ due to insufficient coverage of the overall distribution. Secondly, the synergy of quantization and cache amplifies exposure bias (See Appendix G.1 for a detailed definition) within the sampling distribution, a problem that is less pronounced when exploring either quantization or cache individually, which can be observed in Figure 2. Furthermore, this exposure bias induces an accumulated shift in the variance of denoised outputs as the sampling iterations increase.

To restore the generation capability of models while preserving the acceleration benefits from combining quantization and and cache mechanisms, we tackle the above challenges by developing two essential techniques, constituting our hybrid acceleration mechanism: (i) temporal-aware parallel clustering (TAP) and (ii) distribution variance compensation (VC).

In particular, our TAP aims to restore the reduced sample efficacy in calibration datasets caused by cache operation, thereby enabling more accurate identification and correction of quantization errors. Notably, a naïve method to overcome the reduction of sample efficacy is to increase the dataset size. However, it will introduce excessive redundant data and unnecessary computational costs (see Table 12 for supporting experiments). In contrast, TAP dynamically constructs calibration datasets by efficiently selecting the most informative and distinguishable samples from large-scale datasets through a parallel clustering scheme. Unlike traditional spectral clustering, which suffers from prohibitive computational complexity of $O(n^3)$ (Yan et al., 2009; Li et al., 2011; Chen & Cai, 2011) or $O(n^2)$ with optimized algorithms (Halko et al., 2011; Feng et al., 2018; Martin et al., 2018). TAP integrates temporal sequences with data distribution to enable parallel subsampling of size $r$, thereby reducing the complexity to $O(rn)$, where $r \ll n$. This design leverages the time-sensitive nature of calibration datasets in generative models, as highlighted in recent studies (Li et al., 2023b; 2024b), allowing TAP to achieve effective clustering and sampling that faithfully represent the overall distribution while avoiding excessive redundancy and computational costs.

Our in-depth analysis of the image generation process reveals a strong link between image variance and exposure bias, as shown in Sec.2.2. To address this, we propose the VC, a tailored approach that adaptively mitigates exposure bias through variance correction. Unlike methods that introduce an additional neural network to predict errors in corrupted estimations (Wimbauer et al., 2024), our approach requires no additional training. Instead, it utilizes a small batch of intermediate samples to compute a reconstruction factor, which adaptively corrects feature variance at each timestep. This method effectively reduces exposure bias, resulting in notable improvements in overall model performance.

The contributions of this paper can be summarized as follows:

- We are the first to systematically investigate the combined use of quantization and caching techniques from both theoretical and empirical perspectives, demonstrating the substantial potential of this approach to alleviate computational burdens.

- We identify two critical challenges when integrating quantization and cache: (1) the generation of highly redundant samples in calibration datasets; and (2) the emergence of exposure bias caused by distributional variance shifts in the model's output, which becomes exacerbated over iterations.

- We propose two novel methods: (1) TAP dynamically selects informative and distinct samples from large-scale datasets to optimize calibration dataset efficacy. (2) VC, an adaptive approach that mitigates exposure bias by correcting feature variance at each timestep requiring no additional training.

- Extensive empirical results demonstrate that our approach exhibits strong generalization, being broadly applicable across diverse model architectures, generation tasks, and various combinations of quantization–cache methods and settings, while simultaneously accelerating diffusion-based image generation by up to $12.7\times$ without compromising generative quality.

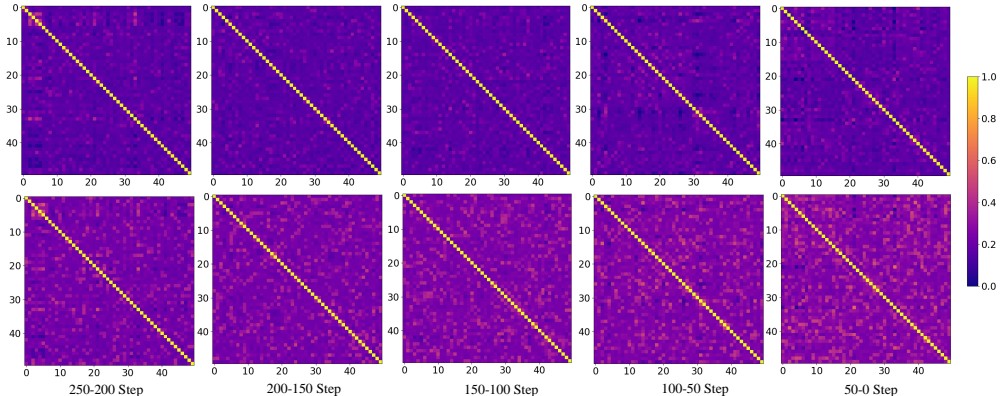

Figure 1: Cosine similarity analysis across time steps in DiT for calibration data. This visualization is based on a 250-step DDIM sampling process. Calibration data were collected both without (up) and with (bottom) cache; samples positioned further to the right represent data closer to the final step $x_0$. The heatmap reveals high similarity in calibration datasets when quantization meets cache, particularly in later diffusion stages. This observation motivates our calibration strategy, highlighting a clear requirement to reduce redundancy and improve efficacy. A more detailed theoretical and experimental analysis is provided in Appendix D.1.

## 2 BACKGROUND AND MOTIVATION

### 2.1 QUANTIZATION AND CACHE

Quantization, a pivotal stage in model deployment, has often been scrutinized for its ability to reduce memory footprints and inference latencies. Typically, its quantizer $Q(X|b)$ is defined as follows:

$$Q(X|b) = \text{clip}(\left\lfloor \frac{X}{s} \right\rceil + z, 0, 2^b - 1) \tag{1}$$

Where $s$ (scale) and $z$ (zero-point) are quantization parameters determined by the lower bound $l$ and the upper bound $u$ of $X$, which are usually defined as follow:

$$l = \min(X), u = \max(X) \tag{2}$$

$$s = \frac{u - l}{2^b - 1}, z = \text{clip}(\left\lfloor -\frac{l}{s} \right\rceil + z, 0, 2^b - 1) \tag{3}$$

Using a calibration dataset and Equation 2and 3, we can derive the statistical information for $s$ and $z$. Previous research (Williams & Aletras, 2024; Lee et al., 2023; Wu et al., 2023; Jaiswal et al., 2023) has examined the performance of downstream tasks across a variety of models, compression methods, and calibration data sources. Their findings indicate that the choice of calibration data can significantly impact the performance of compressed models.

Cache, a technique that leverages the repetitive nature of denoising steps in diffusion models, significantly reduces computational costs while maintaining the quality of generated samples. cache mechanisms operate by storing and reusing intermediate outputs during the sampling process, avoiding the need for redundant calculations at each step. The key parameter in this approach is the cache interval $N$, which dictates how often features are recomputed and cached. Initially, features for all layers are cached, and at each time step $t$, if $mod N = 0$, the model recomputes and updates the cache. For the following $N - 1$ steps, the model reuses these cached features, bypassing the need for repeated full forward passes. This process efficiently reduces computational overhead, particularly in diffusion models, without sacrificing generative quality.

### 2.2 CHALLENGES IN THE SYNERGY OF QUANTIZATION AND CACHE IN EFFICIENT IMAGE GENERATION

The remarkable performance of quantization and cache naturally leads us to consider the significant potential of their combination for enhancing the efficiency of DiTs. To this end, we leverage standard quantization (Nagel et al., 2021) and caching mechanism (Ma et al., 2024b) to conduct an in-depth analysis, through which we identify two critical issues.

**Challenge 1: Amplification of Exposure Bias**
Past research has consistently shown that exposure bias, resulting from the training-inference discrepancy, has a profound impact on text and image generation models (Ranzato et al., 2015; Schmidt, 2019; Rennie et al., 2017; Ning et al., 2023). Due to the presence of exposure bias, it gradually intensifies with the increase in the number of inference sampling steps, becoming a major cause of error accumulation (Li et al., 2023a; Li & van der Schaar, 2023) (See Appendix G.1 for a more detailed definition). To explore this further, we compared the changes in exposure bias under different acceleration methods and were surprised to find that **when quantization meets cache, exposure bias significantly worsens, whereas it does not occur when either quantization or cache is used in isolation, as shown in Figure 2**.

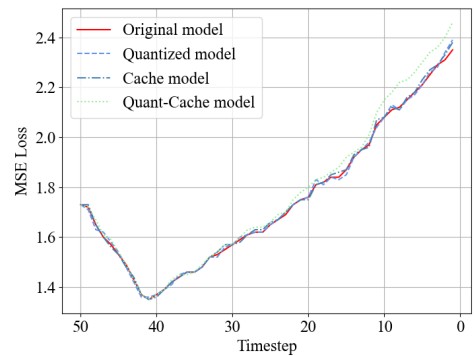

Figure 2: Analysis of exposure bias in DiT models. The mean squared errors between predicted samples and ground truth samples are computed at each time step. While the exposure bias remains relatively stable in both the cached and quantized models compared to the 50-timestep DiT, a noticeable increase in exposure bias is observed when quantization meets cache, leading to accumulation during the generation process.

To analyze the underlying causes, we examined the distributional changes over the generation process using 5,000 images. We observed that this Amplification is due to a change in variance. Specifically, as shown in Figure 3, at the beginning of the denoising process, the span of variance is narrow, and the changes in variance remain stable, fluctuating around 1. As the noise is gradually removed from the white noise, the variance distribution of the ground truth samples spans approximately (0, 0.6), reflecting the diversity of the sample distributions. However, **when considering the synergy of quantization and cache, the distribution shifts to the range of (0.1, 0.7), which aligns closely with the shift trend of exposure bias in Figure 2. We conducted the same experiment for the mean, but no similar phenomenon was observed, detailed analysis can be found in the Appendix I.** This highlights the need to correct variance during the later stages of generation to mitigate its negative impact on exposure bias.

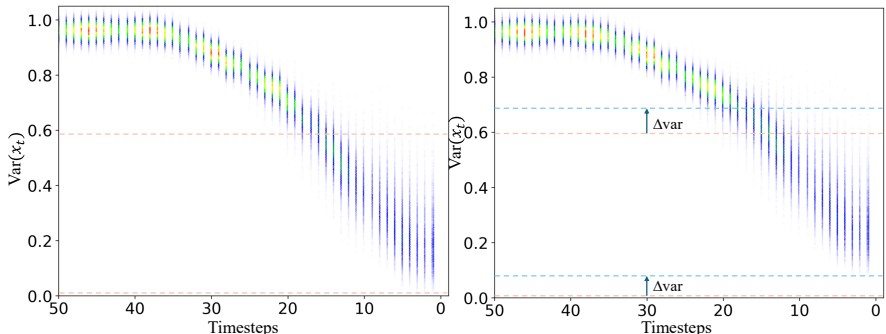

Figure 3: Comparison of the density distribution of the variance of 5000 ImageNet samples across different timesteps. The plots illustrate how the sample distribution variance evolves over time, shown without (left) and with (right) quant-cache. Detailed numerical results are reported in Table 14 in Appendix F. With the incorporation of quant-cache, as denoising progresses, the variance error converges to nearly 100%, and the sample distribution variance gradually deviates toward Gaussian white noise.

**Challenge 2: Degradation in Calibration Dataset Effectiveness** In diffusion quantization, previous works (Liu et al., 2024a; Zhao et al., 2024b; Li et al., 2023b) often randomly sample intermediate inputs uniformly across all time steps to generate a small calibration set. This strategy leverages the smooth transition between consecutive time steps, ensuring that a limited calibration set can still represent the overall distribution effectively (Li et al., 2024b). **However, when quantization meets cache, this balance is disrupted, significantly reducing the effectiveness of the calibration dataset.** In

Appendix D.1, we provide a more comprehensive and detailed theoretical and experimental analysis of this phenomenon.

To visualize this issue, we followed the setup in (Wu et al., 2024) and constructed multiple calibration datasets, each consisting of 250-step samples. We then computed the cosine similarity between these samples and observed a substantial rise in similarity compared to non-cached scenarios (see Figure 1). Furthermore, as the diffusion process approaches the final step $x_0$, sample similarity increases dramatically, with some exceeding 60%. Paradoxically, these later-stage samples are more reliable and valuable for accurate calibration. **This indicates that a large portion of calibration samples, despite their computational cost, do not contribute additional useful information for quantization**, significantly reducing the overall effectiveness of the calibration dataset.

# 3 METHOD

## 3.1 TEMPORAL-AWARE PARALLEL CLUSTERING FOR CALIBRATION

In this section, we present Temporal-Aware Parallel Clustering (TAP), a novel method that integrates both spatial data distribution and temporal dynamics to address clustering challenges in datasets with complex feature interactions and inherent temporal patterns. TAP leverages parallel subsampling to efficiently combine spatial and temporal similarities, providing a robust approach for generating calibrated datasets.

**Algorithm Overview** Given a dataset $T$ with $N$ samples, TAP reduces computational complexity through subsampling, followed by parallel processing across multiple subsampled sets. Each subsample is generated via random sampling, where the probability of selecting a sample is $p_i = \frac{n}{N}$, with $n$ being the number of samples per subsample. By repeating this process, we obtain $m$ subsampled sets $\{S_1, S_2, \ldots, S_m\}$. The parallel subsampling approach offers two key advantages: (1) it mitigates potential random noise and distributional biases within the dataset, and (2) it significantly improves computational efficiency.

For each subsampled set, a similarity matrix $A_{\text{final}}^{(i)}$ is constructed. Next, spectral clustering is applied to each weighted similarity matrix $A_{\text{final}}$ to detect communities. First, we compute the normalized Laplacian matrix for each parallel subsampling set, as follow:

$$L^{(i)} = (D_r^{(i)})^{\frac{1}{2}} A_{\text{final}}^{(i)} (D_c^{(i)})^{\frac{1}{2}} \in \mathbb{R}^{N \times n} \quad (4)$$

Given the subsampled similarity matrix $A_{\text{final}}^{(i)}$, where $D$ is a diagonal matrix with the $i$-th diagonal element being $\sum_h A_{kh}$, for $1 \le k \le N$, the degree matrices for the subsampled node set $S_i$ are defined as:

---

**Algorithm 1:** Temporal-Aware Parallel Clustering (TAP)

**Input:** Dataset $T$ with $N$ samples, samples per subsample $n$
**Output:** Cluster assignments for dataset $T$
**for** $i = 1$ *to* $m$ ***in parallel*** **do**
    Generate subsample $S_i$ from $T$ with $|S_i| = n$;
    Compute $A_{\text{spatial}}^{(i)}$:;
        **for** *each* $(x_k, x_h) \in S_i$ **do**
            $A_{\text{spatial},kh}^{(i)} \leftarrow \frac{x_k \cdot x_h}{\|x_k\|\|x_h\|}$;
    Compute $A_{\text{temporal}}^{(i)}$:;
        **for** *each* $(t_k, t_h) \in S_i$ **do**
            $A_{\text{temporal},kh}^{(i)} \leftarrow \exp(-|t_k - t_h|)$;
    Combine to $A_{\text{final}}^{(i)}$;
    Compute $D_r^{(i)}, D_c^{(i)}$;
    $L^{(i)} \leftarrow (D_r^{(i)})^{1/2} A_{\text{final}}^{(i)} (D_c^{(i)})^{1/2}$;
    Extract the top $k$ eigenvectors of $L^{(i)}$ and perform k-means clustering on the eigenvector matrix rows;
Aggregate cluster assignments from all subsamples to produce final results;

---

$$D_r^{(i)} = \text{diag}\left\{(D_{r,k}^{(i)})_{k=1}^N\right\}, D_c^{(i)} = \text{diag}\left\{(D_{c,h}^{(i)})_{h=1}^N\right\} \quad (5)$$

The top $k$ eigenvectors of $L$ are then extracted, and k-means clustering is performed on the rows of the resulting eigenvector matrix to produce the final clustering results.

As the entire dataset is divided into $k$ categories, we can uniformly sample from these categories to construct the final calibration dataset, ensuring that its data distribution perfectly covers the overall distribution of the original dataset. The detailed algorithm flow is shown in Algorithm 1.

**Definition of Similarity Matrices** $A_{\text{final}}^{(i)}$    Drawing from the prior work (Li et al., 2023b; He et al., 2024), datasets $T$ exhibit complex feature distributions and inherent temporal patterns. To account for these aspects, we construct a comprehensive similarity measure by combining spatial and temporal similarities. Specifically, for each subset $S_i$, we compute the data similarity matrix $A_{\text{data}}^{(i)}$ based on the feature space, and the temporal similarity matrix $A_{\text{time}}^{(i)}$, which captures temporal correlations. We then construct a weighted similarity matrix for each subsample, which combines Spatio-Temporal similarities:

$$A_{\text{final}}^{(i)} = \alpha A_{\text{spatial}}^{(i)} + (1-\alpha) A_{\text{temporal}}^{(i)} \tag{6}$$

where $\alpha$ represents adjustable weights that balance the influence of data spatial and temporal properties.

**Spatial Similarity Matrix** $A_{\text{spatial}}^{(i)}$ captures the similarity between samples in terms of their data features. For each pair of samples $x_k$ and $x_h$ from the subsampled set $S_i$, the element $A_{\text{spatial},kh}^{(i)}$ represents how similar these two samples are based on their feature vectors, which could be defined as:

$$A_{\text{data},kh}^{(i)} = \frac{x_k \cdot x_h}{\|x_k\|\|x_h\|} \tag{7}$$

**Temporal Similarity Matrix** $A_{\text{temporal}}^{(i)}$ captures the similarity between samples based on their temporal relationships. For each pair of samples with timestamps $t_k$ and $t_h$, the element $A_{\text{temporal},kh}^{(i)}$ could be defined as:

$$A_{\text{time},kh}^{(i)} = \exp\left(-|t_k - t_h|\right) \tag{8}$$

## 3.2 Variance Compensation (VC) for Exposure Bias

Assume that a random variable $f$ follows a normal distribution, denoted as $f \sim \mathcal{N}(\mu, \sigma^2)$, where $\mu$ represents the mean and $\sigma^2$ denotes the variance. To alter the variance of $f$, we can apply a scaling transformation. If the objective is to modify the variance to a new value $\sigma_{\text{new}}^2$, the transformation can be defined as follows:

$$Y = \mu + \frac{\sigma_{\text{new}}}{\sigma}(f - \mu) \tag{9}$$

In this formulation, $Y$ will conform to a new normal distribution given by $Y \sim \mathcal{N}(\mu, \sigma_{\text{new}}^2)$, where $\frac{\sigma_{\text{new}}}{\sigma}$ serves as the scaling factor for the variance adjustment. However, in practice, directly determining $\frac{\sigma_{\text{new}}}{\sigma}$ may not be feasible. Consequently, we introduce a timestep-dependent reconstruction scaling factor $\mathbf{K} \in \mathbb{R}^{St \times C}$ within the Intermediate samples $\hat{x}$, where $St$ indicates the number of denoising steps and $C$ signifies the number of channels corresponding to the estimated noise. The reconstructed Intermediate samples $\tilde{x}_t$ at timestep $t$ can thus be represented as follows:

$$\tilde{x}_t = \mu_t + \mathbf{K}_t \cdot (\hat{x}_t - \mu_t) \tag{10}$$

where $(\cdot)$ represents the channel-wise multiplication. Next, we need to select a suitable optimization objective $\mathcal{L}$ to efficiently reconstruct $\tilde{x}_t$.

Inspired by existing work(Finkelstein et al., 2019; Nagel et al., 2019; Yao et al., 2024), we adopt the following objective to more comprehensively measure the discrepancy between the reconstructed feature and the target feature. In this formulation, the inverse root quantization-to-noise ratio (rQNSR) (Finkelstein et al., 2019; Yao et al., 2024) is integrated to enhance the MSE criterion's sensitivity to channel-specific noise effects.

$$\mathbf{K}_t = \underset{\mathbf{K}_t}{\operatorname{argmin}}(\text{rQNSR}(\tilde{x}_t, x_t')^2 + \text{MSE}(\tilde{x}_t, x_t')) \tag{11}$$

Equation.11 transforms the optimization problem into minimizing a function with respect to $\mathbf{K}_t$. By taking the derivative of the function with respect to $\mathbf{K}_t$ and setting the derivative to zero, we can obtain the analytical solution for $\mathbf{K}_t$. In Appendix D.2 and D.4, we provide the theoretical foundation and detailed derivation of VC, from which the final formula is obtained as follow:

$$\mathbf{K}_t = \frac{\sum_n^N (x_{t,n}' - \mu_t)(\hat{x}_{t,n} - \mu_t) + \sum_n^N \frac{\hat{x}_{t,n} - \mu_t}{x_{t,n}'}}{\sum_n^N (\hat{x}_{t,n} - \mu_t)^2 + \sum_n^N \frac{(\hat{x}_{t,n} - \mu_t)^2}{x_{t,n}'^{2+}}} \tag{12}$$

where $N$ denotes the number of samples across the optimization, and the reference $x'$ is obtained during calibration in the quantization process.

# 4 EXPERIMENTS

## 4.1 EXPERIMENTAL SETTINGS

Our experimental setup closely follows the original configuration used in the Diffusion Transformers (DiTs) study (Peebles & Xie, 2023). We evaluate the performance of our method on the ImageNet dataset (Deng et al., 2009) , using pre-trained, class-conditional DiT-XL/2 models (Peebles & Xie, 2023) at image resolutions of both $256 \times 256$ and $512 \times 512$. The DDPM solver (Ho et al., 2020) with 250 sampling steps is employed for the primary generation process, while additional evaluations with reduced sampling steps of 100 and 50 are conducted to further test the robustness of our approach.

For fair benchmarking, all methods utilize uniform quantizers for all activations and weights, with channel-wise quantization for weights and tensor-wise for activations. To create a calibration dataset, we generate large-scale samples across the ImageNet classes during the diffusion process, forming a dataset $D_l$. We utilize the TAP algorithm to select a final set for quantization calibration. Specifically, three parallel sampling processes are performed, with each sampling selecting only 1/20 of the samples. This allows us to split $D_l$ into 100 categories, from which we randomly choose 3-10 samples per category, ultimately forming a set of 800 calibration samples—following the implementation of previous works (Wu et al., 2024). All experiments are conducted on NVIDIA RTX A100 GPUs, and our code is based on PyTorch (Paszke et al., 2019).

To comprehensively assess the quality of generated images, we employ four evaluation metrics: Fréchet Inception Distance (FID) (Heusel et al., 2017), spatial FID (sFID) (Salimans et al., 2016; Nash et al., 2021), Inception Score (IS) (Salimans et al., 2016; Barratt & Sharma, 2018), and Precision. All metrics are computed using the ADM toolkit (Dhariwal & Nichol, 2021). For fair comparison across all methods, including the original models, we sample 10,000 images for ImageNet 256×256 and 5,000 images for ImageNet 512×512, consistent with the standards used in prior studies (Nichol & Dhariwal, 2021; Shang et al., 2023).

## 4.2 COMPARISON ON PERFORMANCE

We conduct a comprehensive evaluation of our method against prevalent baselines, being the first to explore the combined effects of quantization and cache. Our benchmarking includes PTQ4DM (Shang et al., 2023), Q-Diffusion (Li et al., 2023b), PTQD (He et al., 2024), PTQ4DM (Shang et al., 2023) , Learn-to-Cache (Ma et al., 2024a), RepQ (Li et al., 2023c), and Fora (Selvaraju et al., 2024). All quantization methods use uniform quantizers, applying channel-wise quantization to weights and tensor-wise quantization to activations, while cache methods store and reuse outputs from self-attention and MLP layers.

Tables 1a and 1b summarize results for large-scale, class-conditional image generation on ImageNet at resolutions of 256×256 and 512×512 using DiT, demonstrating our method's effectiveness across different datasets, and timestep settings. Notably, under 8-bit quantization, our method closely matches the generative quality of original models while offering substantial computational efficiency.

## 4.3 GENERALITY OF THE METHOD

**Generality Across Diverse Tasks**  We further extend our comparison to other unconditional generation, video generation, and text-conditional generation tasks. Specifically, we conduct experiments with LSUN-Bedroom and LSUN-Church datasets (Yu et al., 2015) on LDM (Rombach et al., 2022), Sora (Liu et al., 2024b) on VBench (Huang et al., 2024),**FLUX.1** (Black-Forest-Labs, 2024) and PixArt-$\Sigma$ (Chen et al., 2024a) on MJHQ-30K (Li et al., 2024a), as well as PartiPrompt (Yu et al., 2022) and MS-COCO (Lin et al., 2014) on Stable Diffusion. The corresponding results are reported in Tables 1c, 8, and 9, 17 in Appendix B and Appendix K.

**Generality Across Methods and Settings**  We construct diverse baselines by randomly combining existing quantization and cache methods to examine the applicability of Q&C under different method compositions. The detailed procedures and results are provided in Table 10. In addition, we investigate the generalization of Q&C under 4-bit and 6-bit settings, with the results summarized in Table 11 in Appendix C.

Table 1: Performance comparison across multiple datasets, where Q&C* denotes results using PTQ4DiT (Wu et al., 2024) with Deepcache (Ma et al., 2024b), Q&C† denotes results using PTQ4DiT (Wu et al., 2024) with learn-to-cache (Ma et al., 2024a), Q&C** denotes APQ-DM (Wang et al., 2023) with Deepcache (Ma et al., 2024b).

(a) ImageNet 256 × 256 with W8A8

| Steps | Method | Speed | FID ↓ | sFID ↓ | IS ↑ | Precision ↑ |
|---|---|---|---|---|---|---|
| 250 | DDPM | 1× | 4.53 | 17.93 | 278.50 | 0.8231 |
| | PTQ4DM | 2× | 21.65 | 99.98 | 134.22 | 0.6342 |
| | Q-Diffusion | 2× | 5.57 | 18.22 | 227.50 | 0.7612 |
| | PTQD | 2× | 5.69 | 18.42 | 224.26 | 0.7594 |
| | RepQ | 2× | 4.51 | 18.01 | 264.68 | 0.8076 |
| | PTQ4DiT | 2× | 4.63 | 17.72 | 274.86 | 0.8299 |
| | FORA | 2.06× | 5.06 | 18.12 | 263.84 | 0.8152 |
| | Q&C* | 3.02× | 4.75 | 18.13 | 265.5 | 0.8210 |
| | Q&C† | 3.12× | 4.68 | 17.84 | 268.65 | 0.8195 |
| 100 | DDPM | 2.5× | 5.00 | 17.65 | 274.78 | 0.8068 |
| | PTQ4DM | 5× | 15.36 | 79.31 | 172.37 | 0.6926 |
| | Q-Diffusion | 5× | 7.93 | 19.46 | 202.84 | 0.7299 |
| | PTQD | 5× | 8.12 | 19.64 | 199.00 | 0.7295 |
| | RepQ | 5× | 5.20 | 19.87 | 254.70 | 0.7929 |
| | PTQ4DiT | 5× | 4.73 | 17.83 | 277.27 | 0.8270 |
| | Q&C* | 6.95× | 4.83 | 18.24 | 265.52 | 0.8056 |
| | Q&C† | 7.23× | 4.75 | 18.02 | 267.96 | 0.8065 |
| 50 | DDPM | 5× | 5.22 | 17.63 | 237.8 | 0.8056 |
| | PTQ4DM | 10× | 17.52 | 84.28 | 154.08 | 0.6574 |
| | Q-Diffusion | 10× | 14.61 | 27.57 | 153.01 | 0.6601 |
| | PTQD | 10× | 15.21 | 27.52 | 151.60 | 0.6578 |
| | RepQ | 10× | 7.17 | 23.67 | 224.83 | 0.7496 |
| | PTQ4DiT | 10× | 5.45 | 19.50 | 250.68 | 0.7882 |
| | Learn-to-Cache | 6.3× | 5.21 | 17.60 | 245.45 | 0.8122 |
| | Q&C* | 12.05× | 5.56 | 19.73 | 248.96 | 0.7812 |
| | Q&C† | 12.7× | 5.43 | 19.52 | 250.68 | 0.7895 |

(b) ImageNet 512 × 512 with W4A8

| Steps | Method | Speed | FID ↓ | sFID ↓ | IS ↑ | Precision ↑ |
|---|---|---|---|---|---|---|
| 100 | DDPM | 1× | 9.06 | 37.58 | 239.03 | 0.8300 |
| | PTQ4DM | 2.5× | 70.63 | 57.73 | 33.82 | 0.4574 |
| | Q-Diff | 2.5× | 62.05 | 57.02 | 29.52 | 0.4786 |
| | PTQD | 2.5× | 81.17 | 66.58 | 35.67 | 0.5166 |
| | RepQ | 2.5× | 62.70 | 73.29 | 31.44 | 0.3606 |
| | PTQ4DiT | 2.5× | 19.00 | 50.71 | 121.35 | 0.7514 |
| | Q&C* | 3.75× | 18.76 | 50.14 | 120.31 | 0.74.95 |
| | Q&C† | 4× | 19.05 | 50.71 | 121.11 | 0.7533 |
| 50 | DDPM | 2× | 11.28 | 41.70 | 213.86 | 0.8100 |
| | PTQ4DM | 5× | 71.69 | 59.10 | 33.77 | 0.4604 |
| | Q-Diff | 5× | 53.49 | 50.27 | 38.99 | 0.5430 |
| | PTQD | 5× | 73.45 | 59.14 | 39.63 | 0.5508 |
| | RepQ | 5× | 65.92 | 74.19 | 30.92 | 0.3542 |
| | PTQ4DiT | 5× | 19.71 | 52.27 | 118.32 | 0.7336 |
| | Q&C* | 6.0× | 20.18 | 52.45 | 117.34 | 0.7285 |
| | Q&C† | 6.5× | 19.71 | 52.26 | 118.45 | 0.7342 |

(c) The Generality with more datasets (W8A8)

| Step | Method | Speed | LSUN-Bedroom | | | LSUN-Church | | |
|---|---|---|---|---|---|---|---|---|
| | | | FID ↓ | sFID ↓ | IS ↑ | FID ↓ | sFID ↓ | IS ↑ |
| 100 | DDIM | 1× | 6.39 | 9.45 | 2.45 | 10.98 | 16.16 | 2.76 |
| | PTQ4DM | 2× | 7.48 | 12.42 | 2.23 | 10.98 | 17.28 | 2.76 |
| | Q-Diff | 2× | 7.04 | 12.24 | 2.27 | 12.72 | 16.96 | 2.72 |
| | APQ-DM | 2× | 6.46 | 11.82 | 2.55 | 9.04 | 16.74 | 2.84 |
| | Q&C** | 3 × | 6.52 | 11.83 | 2.55 | 9.10 | 16.73 | 2.83 |

The experimental findings demonstrate that **Q&C exhibits strong generalization**, being broadly applicable across diverse model architectures, generation tasks, and various combinations of quantization–cache methods and settings.

## 4.4 ABLATION STUDY

**Individual Contributions of TAP and VC** To assess the effectiveness of TAP and VC, we conducted an ablation study using the W8A8 quantization setup on the ImageNet dataset at a resolution of 256 × 256, employing 50 sampling timesteps. We evaluated three method variants: (i) Baseline, which leverages the latest quantization and cache techniques, specifically PTQ4DiT (Wu et al., 2024) combined with Learn-to-Cache (Ma et al., 2024a) on DiTs; (ii) Baseline + TAP, which selects an optimized calibration dataset via TAP; and (iii) Baseline + TAP + VC, incorporating both components. The results, presented in Table 2, demonstrate performance improvements with each added component, underscoring their effectiveness.

Notably, the results reveal that TAP and VC contribute significantly to the quality of generated outputs, indicating that our experiments in Sec.2.2 accurately identified key challenges in the combined use of quantization and cache, and that our methods effectively address these issues. Specifically, the simple stacking of state-of-the-art quantization and cache methods in the baseline led to a sharp drop in generative quality, whereas adding TAP and VC resulted in substantial improvements, reducing FID by 8.24 and sFID by 6.34, significantly outperforming the baseline.

Table 2: Ablation study on ImageNet 256 × 256 for 50 timesteps

| Method | FID ↓ | sFID ↓ | IS ↑ | Precision ↑ | Speed ↑ |
|---|---|---|---|---|---|
| - | 5.22 | 17.63 | 237.8 | 0.8056 | 5x |
| PTQ4DiT | 5.45 | 19.50 | 250.68 | 0.7882 | 10x |
| Baseline | 13.67 | 25.86 | 189.65 | 0.7124 | 11.5x |
| + VC | 9.65 | 22.34 | 210.35 | 0.7445 | 12.1x |
| + TAP | 8.34 | 21.65 | 220.67 | 0.7566 | 12.3x |
| +TAP +VC | 5.43 | 19.52 | 250.68 | 0.7895 | 12.7x |

**Effectiveness of TAP** To demonstrate the superiority of the TAP method, we compare it with several common clustering methods on ImageNet 256 × 256, covering representative algorithms

from partition-based, density-based, and hierarchical clustering approaches. Specifically, we select K-Means (Likas et al., 2003), DBSCAN (Deng, 2020), and Agglomerative (Murtagh & Legendre, 2014) Clustering for comparison. The results are as Tab 4.

Table 3: Ablation for similarity Matrices $\alpha$

| $\alpha$ | FID ↓ | sFID ↓ | IS ↑ | Precision ↑ |
|---|---|---|---|---|
| 0.3 | 5.57 | 19.58 | 248.63 | 0.7823 |
| 0.4 | 5.46 | 19.55 | 250.62 | 0.7863 |
| 0.5 | 5.43 | **19.52** | **250.68** | **0.7895** |
| 0.6 | **5.36** | 19.56 | 249.53 | 0.7875 |
| 0.7 | 5.45 | 19.46 | 248.96 | 0.7725 |

Table 4: TAP with Different Clustering Methods

| Method | FID ↓ | sFID ↓ | IS ↑ | Precision ↑ |
|---|---|---|---|---|
| KMeans | 10.31 | 23.65 | 195.43 | 0.7332 |
| DBSCAN | 10.12 | 23.21 | 201.35 | 0.7365 |
| Agglomerative | 9.56 | 22.13 | 202.12 | 0.7345 |
| TAP (ours) | 8.34 | 21.65 | 220.67 | 0.7566 |

**Hyperparameters in TAP**    TAP leverages spatial data distribution and temporal dynamics to construct similarity matrices. To assess the impact of the parameter $\alpha$ in Equation.6 , we conducted ablation experiments on ImageNet $256 \times 256$, with the results shown in Table 3

**Visualization of VC**    To examine whether the VC effectively mitigates exposure bias, **we provide comprehensive visualizations in the Appendix G.2.**

## 5   DISCUSSION: WHY QAT IS NOT USED AND WHETHER IT SOLVES THE QUANTIZATION–CACHE INTERACTION

Quantization-aware training (QAT) is known to provide higher robustness than post-training quantization (PTQ). However, QAT is not a practical option for diffusion transformer (DiT) models in our target setting, and it does not resolve the core interaction between quantization and cache reuse. We summarize the key reasons below.

**(1) QAT is prohibitively expensive for diffusion models.**    Performing QAT requires end-to-end retraining with simulated quantization throughout the forward pass. For large DiT architectures (e.g., Open-Sora 11B), a single epoch of QAT can cost hundreds of GPU hours. Our goal, in contrast, is to enable a lightweight and scalable quantization pipeline that completes within minutes and applies to any pre-trained model.

**(2) QAT violates the plug-and-play requirement.**    The proposed method is designed to support drop-in deployment, requiring: (i) no access to original training data, (ii) no fine-tuning or retraining pipeline, and (iii) only a small calibration set.

QAT, however, demands access to the full training dataset as well as the full training infrastructure, making it impractical in realistic model deployment scenarios.

**(3) QAT does not address the quantization–cache consistency issue.**    Even with QAT, the numerical stability of cached intermediate features across diffusion steps remains unresolved. QAT optimizes robustness against quantization noise but does not enforce the step-to-step cache consistency required by cache reuse mechanisms. Therefore, QAT is insufficient for stabilizing cached values under diffusion dynamics.

**Experimental Comparison.**    Following prior work Nagel et al. (2021), we conduct QAT on `DiT/XL-2` for ImageNet $256 \times 256$ with 50 diffusion steps. QAT uses the full ImageNet training set and is trained on 8 A100 GPUs. In contrast, our Q&C pipeline requires only one A100 and an 800-sample calibration set. Results are summarized in Table 5.

Table 5: QAT vs. Q&C on DiT/XL-2 for ImageNet $256 \times 256$ (50 steps).

| **Method** | FID↓ | sFID↓ | IS↑ | Precision↑ |
|---|---|---|---|---|
| DDPM (FP) | 5.22 | 17.63 | 237.8 | 0.8056 |
| QAT | 5.67 | 18.42 | 225.58 | 0.7565 |
| QAT + DeepCache | 8.95 | 23.23 | 200.15 | 0.7351 |

The results show that QAT alone does not recover full-precision performance, and combining QAT with caching leads to further degradation due to unresolved cache–quantization inconsistency. These findings indicate that a PTQ-aware cache mechanism remains necessary even in the presence of QAT.

## 6 RELATED WORK

Enhancing the efficiency of diffusion models has become increasingly necessary due to the high computational cost associated with larger models Gao et al. (2025); Ding et al. (2025); Lu et al. (2025); Xie et al. (2025); Li et al. (2025). Quantization and cache mechanisms offer promising approaches to improve the computational efficiency of diffusion models.

**Quantization** methods such as Post-Training Quantization (PTQ) Ding et al. (2023); Wu et al. (2023) have gained attention for their ability to reduce model size and inference time without requiring retraining, making them computationally efficient. Unlike Quantization-Aware Training (QAT), PTQ only requires minimal calibration and can be implemented in a data-free manner by generating calibration datasets using the full-precision model. Techniques like Q-Diffusion (Li et al., 2023b) apply PTQ methods proposed by BRECQ (Li et al., 2021) to optimize performance across various datasets, while PTQD (He et al., 2024) mitigates quantization errors by integrating them with diffusion noise. More recent work, such as EfficientDM (He et al., 2023), fine-tunes quantized diffusion models using QALoRA (Xu et al., 2023; Han et al., 2024), while HQ-DiT (Liu & Zhang, 2024) adopts low-precision floating-point formats, utilizing data distribution analysis and random Hadamard transforms to reduce outliers and enhance quantization performance with minimal computational cost.

**Cache** aims to mitigate the computational redundancy in diffusion model inference by leveraging the repetitive nature of sequential diffusion steps. cache in diffusion models leverages the minimal change in high-level features across consecutive steps, enabling reuse of these features while updating only the low-level details. For instance, studies (Ma et al., 2024b; Wimbauer et al., 2024) reuse feature maps from specific components within U-Net architectures, while (Hunter et al., 2023) focuses on reusing attention maps. Further refinements by (Wimbauer et al., 2024; So et al., 2023; Ma et al., 2024b) involve adaptive lifetimes for cached features and adjusting scaling to maximize reuse efficiency. Additionally, (Zhang et al., 2024) identifies redundancy in cross-attention during fidelity improvement steps, which can be cached to reduce computation.

Previous research has accumulated extensive work in both quantization and caching. However, there has been little exploration into how these two acceleration mechanisms can be combined effectively and the challenges that arise from their integration. This work aims to identify these challenges and address them systematically.

## 7 CONCLUSION

In this paper, we systematically investigated the impact of integrating quantization techniques with caching mechanisms in efficient generative models, from both theoretical and empirical perspectives. Our study identified key challenges arising when quantization is applied in conjunction with cache, particularly redundancy in calibration datasets and the exacerbation of exposure bias. To address these issues, we proposed two complementary techniques: **Temporal-Aware Parallel Clustering (TAP)** to enhance calibration efficacy, and a **Variance Compensation (VC)** strategy to mitigate exposure bias. Experimental results demonstrate that the combination of TAP and VC substantially improves generation quality while preserving computational efficiency. Furthermore, our findings show that the proposed Q&C framework **exhibits strong generalization**, being broadly applicable across diverse model architectures, generation tasks, and various quantization–cache configurations. We believe that this work paves the way for more efficient and effective generative pipelines.

ACKNOWLEDGEMENT

This work was supported in part by NSFC under Grant 62371434 and U25B2010, the Postdoctoral Fellowship Program of CPSF under Grant Number GZC20252293, the China Postdoctoral Science Foundation-Anhui Joint Support Program under Grant Number 2024T017AH, China Postdoctoral Science Foundation under Grant Number 2025M783529, Anhui Postdoctoral Scientific Research

Program Foundation (No.2025A1015), the Fundamental Research Funds for the Central Universities (No. WK2100250064), ZGCA Project-C20250302.

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

## A  LARGE LANGUAGE MODELS (LLMS) IN PAPER WRITING

In this work, Large Language Models (LLMs) were only employed for proofreading, spelling, and formatting checks. LLMs did not contribute to any aspect of research idea generation, experimental design, or coding.

## B  THE COMPARISON WITH CONCURRENT WORK

We provide a systematic comparison between our Q&C method and two representative contemporaneous works—QuantCache (Wu et al., 2025) and CacheQuant (Liu et al., 2025)—from three perspectives: problem formulation, technical mechanisms, and empirical performance.

Moreover, the contemporaneous works **did not effectively analyze the fundamental issues** arising from the combination of quantization and caching. Instead, they adjusted the system dynamically to achieve a better integration. In contrast, Q&C not only proposes innovative methods to improve the synergy between quantization and caching but also conducts extensive experiments to **analyze the intrinsic phenomena of their interaction**.

Table 6: Differences in Problem Formulation Compared with Concurrent Works

| Comparison | QuantCache | CacheQuant | Q&C |
|---|---|---|---|
| objectives | Accelerate DiTs in Video Generation Tasks | Accelerate diffusion models in image generation tasks | **Focus on the collaborative conflict problem of Q&C itself, and the first mechanism to identify and deal with its linkage error** |
| Application scope | Video-level inference | UNet and DiT | 🔥Broadly applicable to any task built upon diffusion-based architectures. |

Table 7: Differences in Technical Mechanisms Compared with Concurrent Works

| Comparison | QuantCache | CacheQuant | Q&C |
|---|---|---|---|
| Exposure Bias Modeling | ❌Not addressed | ⚠️Indirectly mitigated via error correction | ✅**Explicitly modeled, with VC correcting distribution shift** |
| Quantization-Cache Synergy | ❌Focused only on redundancy | ✅Modeled via dynamic programming | ✅Optimizes calibration quality via TAP, lightweight and direct |
| Training / Scheduling Dependency | Requires retraining or pruning stats | Training-free but needs path scheduling | ✅Fully training-free, no scheduling or model changes |
| Core Advantage | Reduces cache redundancy only | Complex scheduling, limited flexibility | VC: 🔥first training-free drift correction; TAP: boosts robustness & generality |

# C  THE ANALYSIS OF GENERALIZATION

**The Generalization on Different Quantization–Cache Combinations**  To further demonstrate the **generality and robustness** of our method across different quantization strategies and cache designs, we include six additional baselines and conduct comparisons on both the **ImageNet-256** and **LSUN-Bedroom** datasets:

- **Base1**: Standard Quantization (Nagel et al., 2021) + DeepCache (Ma et al., 2024b)
- **Base2**: PTQ4DiT (Wu et al., 2024) + Learn-to-Cache (Ma et al., 2024a)
- **Base3**: APQ-DM (Wang et al., 2023) + DeepCache (Ma et al., 2024b)
- **Base4**: PTQ4DiT (Wu et al., 2024) + DeepCache (Ma et al., 2024b)
- **Base5**: PTQ4DiT (Wu et al., 2024) + FORA (Selvaraju et al., 2024)
- **Base6**: Q-Diff (Chen et al., 2025) + DeepCache (Ma et al., 2024b)

**The Generalization on Different Quantization Setting**  To further demonstrate the robustness and generality of our method, we validate Q&C using Q-DiT (Chen et al., 2025) combined with Learn-to-Cache (Ma et al., 2024a) under 4-bit and 6-bit settings. The experimental results are presented in Table 11.

All experiments were conducted on a single NVIDIA A100 GPU. Note that for the W6A8 setting, due to the lack of dedicated hardware acceleration operators, we only provide a theoretical acceleration ratio.

# D  THEORETICAL FOUNDATIONS OF Q&C

## D.1  A MORE DETAILED THEORETICAL ANALYSIS OF THE CALIBRATION DATASET DEGRADATION

**1. Loss of Diversity Due to Caching**  In traditional PTQ setups, intermediate features are sampled uniformly from the full diffusion process:

$$D_{\text{calib}} = \{x_t = \epsilon_\theta(x_0, t) \mid t \sim \mathcal{U}(1, T)\}, \tag{13}$$

which leverages the smooth transition across steps to ensure coverage of diverse feature distributions.

However, when caching is used, the calibration set is reduced to features from a few fixed timesteps:

$$D_{\text{cache}} = \{x_t \mid t \in T_{\text{cache}}, |T_{\text{cache}}| \ll T\}. \tag{14}$$

Table 8: Performance Comparison with CacheQuant on Text-Conditional Generation Tasks

| Datasets | Image256 | | | MS-COCO | | PartiPrompts | |
|---|---|---|---|---|---|---|---|
| Metric | FID↓ | IS↑ | Speed | IS ↑ | Speed | FID↓ | Speed |
| DiT/LDM | 6.02 | 246.24 | 1x | 41.02 | 1x | 27.23 | 1x |
| CacheQuant | 12.42 | 173.17 | 3.08x | 39.41 | 5.18x | 27.15 | 5.2x |
| Q&C | **6.23** | **244.68** | **3.22x** | **40.95** | **5.53x** | **26.35** | **5.3x** |

Table 9: Performance Comparison with Quantcache on Video-generation task

| | Motion | BG. | Subject | Aesthetic | Imaging | Dynamic | Scene |
|---|---|---|---|---|---|---|---|
| Open-Sora | 98.42 | 96.44 | 95.20 | 60.07 | 59.66 | 33.33 | 41.72 |
| Q-DiT | 95.72 | 95.01 | 91.68 | 58.68 | 56.54 | 38.88 | 34.06 |
| PTQ4DiT | 98.02 | 96.33 | 96.23 | 58.40 | 53.29 | 37.50 | 36.36 |
| QuantCache | 98.52 | 96.12 | 94.62 | 58.57 | 55.94 | 31.94 | 36.92 |
| Q&C | **98.62** | **96.34** | **95.10** | **59.15** | **57.33** | **32.34** | **38.52** |

Effectively, the calibration distribution becomes concentrated:

$$p_{\text{cache}}(x_t) \approx \sum_{t \in T_{\text{cache}}} \delta(x_t). \tag{15}$$

We compute cosine similarity between features:

$$\text{Sim}(x_i, x_j) = \frac{\langle x_i, x_j \rangle}{\|x_i\| \cdot \|x_j\|}. \tag{16}$$

Caching increases $\mathbb{E}[\text{Sim}]$, indicating high redundancy and reduced diversity.

**2. Redundancy in Late-Stage Features**    As $t \to 0$ (closer to $x_0$), sample variance decreases:

$$\lim_{t \to 0} \text{Var}(x_t) \to 0 \quad \Rightarrow \quad \text{Sim}(x_i, x_j) \to 1. \tag{17}$$

We observe that as the diffusion proceeds towards denoised outputs (closer to $x_0$), the sample similarity increases dramatically. In cached settings, late-stage representations are more reliable calibration dataset. However, despite their higher reliability, the redundancy among these samples grows, with similarity exceeding 60% in some cases. This saturation effect means that many cached samples are nearly identical, and thus fail to provide new calibration signal.

**3. Inefficient Use of Calibration Budget**    A fixed calibration budget (e.g., 250 samples) is inefficient when many samples are near-duplicates. The quantization objective:

$$\min_{Q} \mathbb{E}_{x \sim D_{\text{calib}}} \left[ \|Q(x) - x\|^2 \right] \tag{18}$$

fails to capture error in underrepresented regions, especially early/mid diffusion steps.

**Summary**    Caching introduces redundancy and distributional bias in the calibration dataset:

- Diversity is reduced by over-sampling fixed steps
- Late-stage features dominate but offer limited marginal gain
- Calibration becomes less representative, hurting quantization accuracy

This raises the following question: ***Can increasing calibration samples alone resolve the issue?***    To evaluate this, we followed the same setup as PTQ4DiT: 25 timesteps $\times 32$ samples for $256 \times 256$ resolution (total 800 samples).

We conducted experiments using DiT-XL/2 (50 steps) on ImageNet with PTQ4DiT+Learn-to-Cache (the current state-of-the-art quantization and caching method), increasing the sample count to 1600, 3200, and 6400.

Even with 6400 samples—requiring over 2.5 days of computation on single A100—the performance still lagged behind our proposed Q&C method. This highlights that increasing sample size alone is insufficient, and our method achieves better results with far less cost, due to its efficient calibration.

Table 10: Performance under Different Quantization–Cache Combinations, Here, $5\times$ denotes a $5\times$ speedup, i.e., using only 50 steps compared to the standard 250-step setting.

| | ImageNet 256 | | | | | | LSUN-Bed | | | |
|---|---|---|---|---|---|---|---|---|---|---|
| Model | FID↓ | sFID↓ | IS↑ | Pre↑ | Speed↑ | Model | FID↓ | sFID↓ | IS↑ | Speed↑ |
| DiT | 5.22 | 17.63 | 237.8 | 0.8056 | $5\times$ | LDM | 6.39 | 9.45 | 2.45 | $1\times$ |
| Base1 | 28.34 | 102.25 | 113.34 | 0.6023 | $12.1\times$ | Base1 | 15.56 | 23.32 | 1.85 | $3\times$ |
| **Base1+Q&C** | **14.35** | **26.55** | **190.89** | **0.7082** | $12.1\times$ | **Base1+Q&C** | **10.21** | **15.32** | **2.15** | $3\times$ |
| Base2 | 13.67 | 25.86 | 189.65 | 0.7124 | $12.5\times$ | Base3 | 11.38 | 16.55 | 1.96 | $3\times$ |
| **Base2+Q&C** | **5.43** | **19.52** | **250.68** | **0.7895** | $12.7\times$ | **Base3+Q&C** | **6.52** | **11.83** | **2.55** | $3\times$ |
| Base4 | 14.56 | 27.71 | 188.63 | 0.7066 | $12.1\times$ | Base6 | 12.87 | 17.94 | 1.95 | $3\times$ |
| **Base4+Q&C** | **5.56** | **19.73** | **248.96** | **0.7812** | $12.05\times$ | **Base6 + Q&C** | **7.01** | **12.25** | **2.31** | $3\times$ |
| Base5 | 15.97 | 27.62 | 175.35 | 0.7054 | $12.2\times$ | | | | | |
| **Base5+Q&C** | **5.12** | **18.33** | **265.55** | **0.8149** | $12.3\times$ | | | | | |

Table 11: Performance under Different Quantization–Cache Combinations

| W/A | Method | FID↓ | sFID ↓ | IS ↑ | Precision | Speed ↑ |
|---|---|---|---|---|---|---|
| 16/16 | FP | 12.40 | 19.11 | 116.68 | 0.6605 | $1\times$ |
| 6/8 | Q-DIT | 12.21 | 18.48 | 117.75 | 0.6631 | $2.6\times$ |
| 6/8 | Q-DIT+learn-to-cache | 15.56 | 21.65 | 100.56 | 0.6113 | $3.58\times$ |
| 6/8 | Q&C | **12.32** | **18.56** | **116.56** | **0.6559** | $3.77\times$ |
| 4/8 | Q-DIT | 15.76 | 19.84 | 98.78 | 0.6395 | $4\times$ |
| 4/8 | Q-DIT+learn-to-cache | 18.95 | 22.34 | 85.92 | 0.5685 | $5.52\times$ |
| 4/8 | Q&C | **16.12** | **20.05** | **99.52** | **0.6385** | $5.79\times$ |

## D.2 THEORETICAL OF VC FOR MITIGATING EXPOSURE BIAS

Let the intermediate feature representation at diffusion step $t$ be a random variable $X_t$, whose distribution during calibration is

$$X_t \sim \mathcal{N}(\mu_t, \sigma_t^2), \tag{19}$$

where $\mu_t$ is the mean and $\sigma_t^2$ is the variance.

During actual sampling, due to quantization errors and caching mechanisms, the feature distribution shifts to

$$\hat{X}_t \sim \mathcal{N}(\hat{\mu}_t, \hat{\sigma}_t^2). \tag{20}$$

The distributional drift that causes exposure bias can be written as:

$$\Delta\mu_t = \hat{\mu}_t - \mu_t, \quad \Delta\sigma_t^2 = \hat{\sigma}_t^2 - \sigma_t^2. \tag{21}$$

where $\Delta\mu_t$ and $\Delta\sigma_t^2$ respectively capture the mean shift and variance shift at timestep $t$. Extensive experiments in Appendix I and Section 2.2 show that the mean drift is generally negligible:

$$|\Delta\mu_t| \approx 0, \tag{22}$$

while the variance drift is significant:

$$|\Delta\sigma_t^2| \gg |\Delta\mu_t|. \tag{23}$$

The diffusion process heavily relies on the variance of features to regulate uncertainty and noise scale. Variance drift causes inaccurate noise estimation, which accumulates errors step-by-step, leading to exposure bias. Specifically:

1.if variance shrinks, the model becomes overconfident and reduces sample diversity;

2.if variance expands, excessive noise destabilizes generation.

To address this, we introduce Variance Correction (VC) by defining a correction factor:

$$\mathbf{K}_t = \frac{\sigma_t}{\hat{\sigma}_t}, \tag{24}$$

and rescale the sampled feature as

$$X_t^{corr} = \mathbf{K}_t \cdot (\hat{X}_t - \hat{\mu}_t) + \mu_t. \tag{25}$$

Table 12: Impact of Increasing Calibration Samples

| Model | Num of Sample | FID↓ | sFID↓ | IS↑ | Precision↑ | Time(h) (single A100) |
|---|---|---|---|---|---|---|
| DiT-XL/2-256 | | 5.22 | 17.63 | 237.8 | 0.8056 | - |
| | 800 | 13.67 | 25.86 | 189.65 | 0.7124 | 7.97 |
| | 1600 | 13.52 | 25.67 | 191.34 | 0.7152 | 16.85 |
| | 3200 | 11.26 | 24.42 | 200.25 | 0.7226 | 32.62 |
| | 6400 | 8.52 | 21.25 | 209.62 | 0.7345 | 64.34 |
| | Q&C(800) | **5.43** | **19.52** | **250.68** | **0.7895** | **8.18** |

This operation restores the expected variance of the feature distribution, mitigating the adverse effects of variance drift. As a result, it significantly reduces error accumulation, enhances sampling stability, and improves overall generation quality.

In summary, variance correction offers a lightweight, training-free, and effective mechanism to compensate for the dominant form of distribution shift—variance drift—thereby mitigating exposure bias without introducing additional model complexity.

### D.3 THEORETICAL OF SYNERGY BETWEEN TAP AND VC

Let $X_t$ denote the feature distribution at timestep $t$ and let $Z$ be the set of calibration samples.

- **VC estimates a correction term**

$$\Delta\sigma = \sigma_{\text{inference}} - \sigma_Z.$$

- The effectiveness of VC depends on the accuracy of $\sigma_Z$ in approximating $\sigma_{\text{inference}}$.
- TAP enhances this approximation by minimizing the distributional shift

$$\left\|\mathbb{E}_{t\sim\mathcal{T}}[\sigma_{Z_t}] - \sigma_{\text{inference}}\right\|_2$$

across all relevant timesteps $\mathcal{T}$.

Thus, TAP indirectly **reduces the correction magnitude** needed by VC and **stabilizes the estimation** of variance across channels.

### D.4 THE DETAILED DERIVATION OF THE TRANSFORMATION PARAMETER $\mathbf{K}_t$

We define the optimal transformation parameter $\mathbf{K}_t$ by minimizing a combined objective of the relative quadratic normalized squared residual (rQNSR)(Finkelstein et al., 2019; Yao et al., 2024) and mean squared error (MSE) between the transformed prediction $\tilde{x}_t$ and the reference $x'_t$:

$$\mathbf{K}_t = \underset{\mathbf{K}_t}{\arg\min}\left(\text{rQNSR}(\tilde{x}_t, x'_t)^2 + \text{MSE}(\tilde{x}_t, x'_t)\right) \tag{26}$$

where $N$ denotes the number of samples used during optimization, $\tilde{x}_t$ is defined as the affine-transformed prediction:

$$\tilde{x}_{t,n} = \mathbf{K}_t(\hat{x}_{t,n} - \mu_t) + \mu_t$$

Substituting equation 26 into the expanded objective, we obtain the loss function:

$$\mathcal{L}(\mathbf{K}_t; \hat{x}_{t,n}, x'_{t,n}) = \frac{1}{N}\sum_{n=1}^{N}\left(\mathbf{K}_t(\hat{x}_{t,n} - \mu_t) + \mu_t - x'_{t,n}\right)^2$$
$$+ \frac{1}{N}\sum_{n=1}^{N}\left(\frac{\mathbf{K}_t(\hat{x}_{t,n} - \mu_t) + \mu_t - x'_{t,n}}{x'_{t,n}}\right)^2 \tag{27}$$

Expanding the terms gives:

$$\mathcal{L}(\mathbf{K}_t) = \frac{1}{N}\sum_{n=1}^{N}(\hat{x}_{t,n}-\mu_t)^2 \cdot \mathbf{K}_t^2 + \frac{1}{N}\sum_{n=1}^{N}\frac{(\hat{x}_{t,n}-\mu_t)^2}{(x'_{t,n})^2}\cdot \mathbf{K}_t^2$$

$$-\frac{2}{N}\sum_{n=1}^{N}(\hat{x}_{t,n}-\mu_t)(x'_{t,n}-\mu_t)\cdot \mathbf{K}_t$$

$$-\frac{2}{N}\sum_{n=1}^{N}\frac{(\hat{x}_{t,n}-\mu_t)(x'_{t,n}-\mu_t)}{x'_{t,n}}\cdot \mathbf{K}_t$$

$$+\frac{1}{N}\sum_{n=1}^{N}(x'_{t,n}-\mu_t)^2 + \frac{1}{N}\sum_{n=1}^{N}\frac{(x'_{t,n}-\mu_t)^2}{(x'_{t,n})^2} \tag{28}$$

Since this is a convex quadratic function with respect to $\mathbf{K}_t$, the optimal solution can be obtained by taking the derivative and setting it to zero:

$$\frac{\partial\mathcal{L}(\mathbf{K}_t)}{\partial\mathbf{K}_t} = 0 \tag{29}$$

Solving equation 29 leads to the following closed-form solution:

$$\mathbf{K}_t = \frac{\sum_{n=1}^{N}(x'_{t,n}-\mu_t)(\hat{x}_{t,n}-\mu_t)+\sum_{n=1}^{N}\frac{(\hat{x}_{t,n}-\mu_t)}{x'_{t,n}}}{\sum_{n=1}^{N}(\hat{x}_{t,n}-\mu_t)^2+\sum_{n=1}^{N}\frac{(\hat{x}_{t,n}-\mu_t)^2}{(x'_{t,n})^2}} \tag{30}$$

## E    RUNTIME OVERHEAD OF TAP

We conducted experiments on a single A100 GPU using PTQ4DiT as an example, with the DiT-XL/2-256 model under sampling steps of 50, and 100. We report the runtime breakdown to show the actual cost of each component. The comparison reveals that TAP brings substantial improvements while having minimal impact on the overall runtime—it accounts for only a small fraction (just 2%-3%) of the total quantization time.

Table 13: Runtime overhead of TAP

| Method | Calibration weight$_{quant\_init}$ | Activation$_{quant\_init}$ | Recon$_{quant}$ | Overhead$_{TAP}$ |
|---|---|---|---|---|
| PTQ4DiT_50 | 14.75 | 16.57 | 13.67 | 433.23 | 0 |
| Q&C | 27.6 | 16.57 | 13.67 | 433.23 | **2.6%** |
| PTQ4DiT_100 | 25.3 | 16.57 | 13.67 | 433.23 | 0 |
| Q&C | 40.4 | 16.57 | 13.67 | 433.23 | **2.9%** |

## F    EVOLUTION OF SAMPLE VARIANCE WITH QUANT-CACHE

We provide a table of numerical changes to more clearly illustrate this effect. As denoising progresses, we observe that the **variance error approaches nearly 100%**, highlighting the significant impact of the quantization-cache interaction.

Table 14: Runtime overhead of TAP

| Variance | 50 | 40 | 30 | 20 | 10 | 0 |
|---|---|---|---|---|---|---|
| DiT | 0.98 | 0.95 | 0.84 | 0.58 | 0.38 | **0.21** |
| Quantization (Nagel et al., 2021) + cache (Ma et al., 2024b) | 0.98 | 0.96 | 0.92 | 0.71 | 0.52 | **0.38** |

# G    ANALYSIS OF EXPOSURE BIAS IN DiT MODELS WITH OUR METHODS.

## G.1    BACKGROUND OF EXPOSURE BIAS

Exposure bias is a well-known issue observed in generative models, such as diffusion models and autoregressive text generation models. This bias stems from discrepancies between training and inference phases, which can lead to significant performance degradation during sampling.

When comparing the training and inference processes in diffusion models, such as in line 4 of Algorithm 2 and line 8 of Algorithm 3, a fundamental difference emerges. During training, diffusion models use $\epsilon_\theta(x_t, t)$,where $x_t$ is a ground truth sample, to predict the noise. However, during inference $\epsilon_\theta(\hat{x}_t, t)$ is deployed, where $\hat{x}_t$ is derived from the model's output at the prior step $t + 1$. this lead to a discrepancy known as traingin-inference mismatch, where training relies on ground-truth sequences, while inference conditions on previously generated outputs (Ranzato et al., 2015; Schmidt, 2019; Rennie et al., 2017; Ning et al., 2023).

---

**Algorithm 2:** DDPM Training

**repeat**
  $x_0 \sim q(x_0)$
  $t \sim \text{Uniform}(\{1, \ldots, T\})$
  $\epsilon \sim \mathcal{N}(\mathbf{0}, \mathbf{I})$ Compute $x_t$
  Take a gradient descent step on
    $\nabla_\theta \| \epsilon - \epsilon_\theta(x_t, t) \|^2$
**until**
converged

---

**Algorithm 3:** DDPM Sampling

Sample $\hat{x}_T \sim \mathcal{N}(\mathbf{0}, \mathbf{I})$
**for** $t = T$ **to** 1 **do**
  **if** $t > 1$ **then**
    Sample $z \sim \mathcal{N}(\mathbf{0}, \mathbf{I})$
  **else**
    Set $z = \mathbf{0}$
  $\hat{x}_{t-1} =$
    $\frac{1}{\sqrt{\alpha_t}} \left( \hat{x}_t - \frac{1-\alpha_t}{\sqrt{1-\bar{\alpha}_t}} \epsilon_\theta(\hat{x}_t, t) \right) + \sigma_t z$
**return** $\hat{x}_0$

---

## G.2    VISUALIZATION OF METHOD EFFECTIVE

**Visualization of Exposure Bias in DiT Models Using VC**    We conducted experiments on exposure bias using our methods and found that our approach significantly mitigates the exacerbation of exposure bias caused by the combined use of quantization and cache mechanisms.

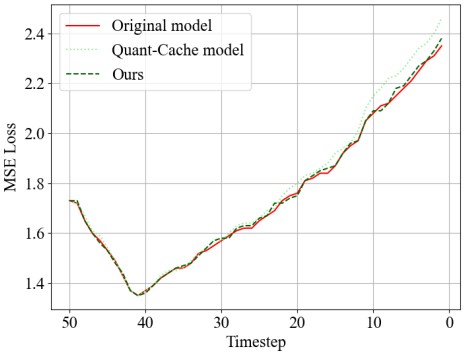

Figure 4: Analysis of exposure bias in DiT models with our methods.

**Visualization of Sample Efficacy in the Calibration Dataset for DiT Models Using TAP**    To demonstrate the improvement of the quantization calibration dataset after applying TAP, we conducted the following visualization experiment on DiT: First, we followed the current SOTA PTQ4DiT calibration dataset generation method to compute the cosine similarity matrices under both cached

and non-cached settings. Then, we applied TAP and computed the improved cosine similarity matrix. It can be observed that the effectiveness of the calibration dataset is improved after applying TAP.

## H EXPERIMENTAL EVALUATION OF THE EFFECTIVENESS OF SIMILARITY MATRICES $A_{\text{FINAL}}^{(i)}$

TAP leverages spatial data distribution and temporal dynamics to construct similarity matrices. To assess the impact of the Spatial Similarity Matrix $A_{\text{spatial}}^{(i)}$ and Temporal Similarity Matrix $A_{\text{tem}}^{(i)}$, we conducted ablation experiments, with the results presented in Table. 15. This fully demonstrates the superiority of our similarity matrices design.

Table 15: Ablation study on ImageNet $256 \times 256$ for similarity Matrices

| method | FID ↓ | sFID ↓ | IS ↑ | Precision ↑ |
|---|---|---|---|---|
| Baseline | 13.67 | 25.86 | 189.65 | 0.7124 |
| w/ $A_{\text{spatial}}^{(i)}$ w/o $A_{\text{temporal}}^{(i)}$ | 5.89 | 20.67 | 245.63 | 0.7713 |
| w/ $A_{\text{spatial}}^{(i)}$ w/ $A_{\text{temporal}}^{(i)}$ | 5.43 | 19.52 | 250.68 | 0.7895 |

## I SUPPLEMENTARY EXPERIMENT FOR CHALLENGE 2: ANALYSIS OF MEAN DISTRIBUTION ACROSS THE GENERATION PROCESS.

To comprehensively analyze the distributional changes over the generation process, we also examined the change in the mean of the sample distribution. As shown in Figure 5, regardless of the method used, at the beginning of the denoising process, the span of variance is narrow, and the changes in mean remain stable, fluctuating around 0. As the noise is gradually removed from the white noise, the variance distribution of the ground truth samples spans approximately (0, 0.6). This indicates that the mean does not exhibit significant shifts during the process and highlights the need to correct variance during the later stages of generation to mitigate its negative impact on exposure bias.

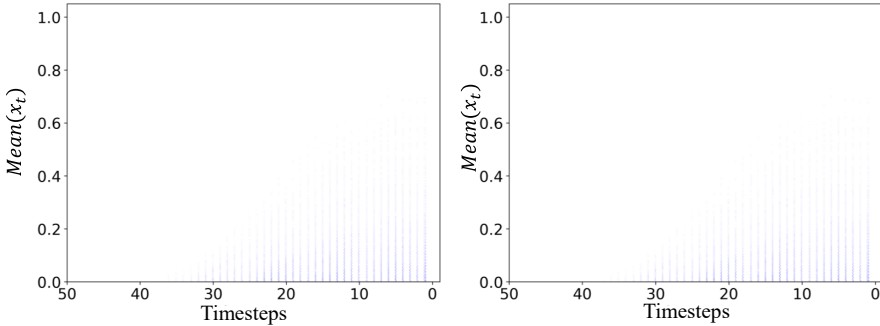

Figure 5: Comparision of the density distribution of the mean of 5000 samples from Imagenet across difference time steps. They illustrate the change in sample distribution mean at various time steps, shown for case without (left) and with (right) quant-cache. As the diffusion progresses, the variance of sample distribution starts to deviate towards Gassian white noise.

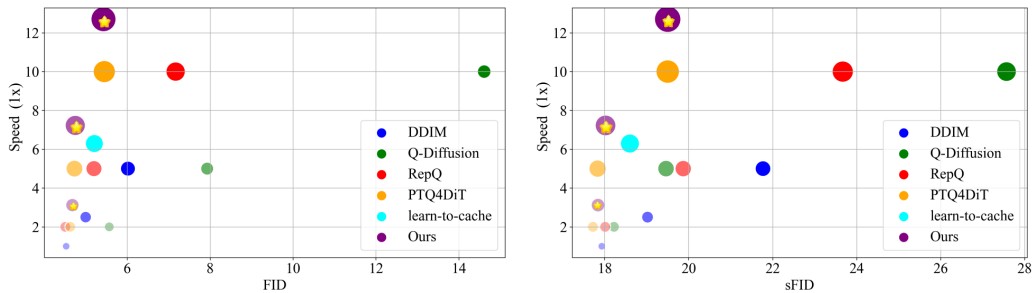

Figure 6: Efficiency-versus-efficacy trade-off across different settings. Bubble size represents the ratio of relative speed-up to generative quality compared to the DDPM baseline at 250 timesteps. We compare various methods in terms of FID (left) and sFID (right) performance across 50, 100, and 250 timesteps. Our method consistently appears in the upper-left region across all settings, achieving maximum acceleration while preserving generative quality.

Figure 6 in the appendix further illustrates the efficiency–efficacy trade-off across different configurations, showing that our approach achieves performance comparable to original models (e.g., 250 timesteps, DDPM) but with markedly reduced computational costs (up to a $12.7\times$ improvement), providing a practical solution for high-quality image generation. Across all tested settings, our method consistently occupies the upper-left region of the performance–efficiency space, surpassing mainstream alternatives and reinforcing its effectiveness and adaptability.

## J  SPEED-UP COMPUTATION AND NON-MULTIPLICATIVE EFFECTS.

The overall acceleration reported in our paper is measured *end-to-end* relative to the FP16 250-step diffusion baseline without cache reuse. Accordingly, the final acceleration integrates the contributions from (i) diffusion step reduction, (ii) PTQ4DiT quantization, and (iii) cache reuse:

$$\text{Total Speed-up} = (\text{Step Reduction Speed-up}) \times (\text{Quantization Speed-up}) \times (\text{Cache Speed-up}).$$

Thus, the reported $12.7\times$ speed-up already reflects the combined improvements of all three components.

**Measurement Setup.**  For clarity and reproducibility, all latency measurements follow the configuration below:

- **Hardware:** NVIDIA A100-SXM-80GB
- **Batch size:** 1
- **Resolution:** $256\times256$
- **Latency metric:** end-to-end
- **Framework:** PyTorch 2.3 with CUDA Graphs

The measured latencies are summarized in Table 16.

| Model | FP16 | W8A8 | W4A8 | Learn-to-Cache |
|---|---|---|---|---|
| Latency (%) | 100 | 51 | 39 | 78 |

Table 16: Measured latency ratios under the standard evaluation setup.

Correspondingly, the quantization-only speed-ups are:

$$\text{W8A8: } 1.96\times, \quad \text{W4A8: } 2.56\times, \quad \text{Learn-to-Cache: } 1.28\times.$$

## K GENERALIZATION TO FLUX AND PIXART-Σ.

To further demonstrate the **strong generalization ability** of Q&C, we additionally evaluate the method on two recent high-quality diffusion models: **FLUX.1** (Black-Forest-Labs, 2024) and **PixArt-**Σ (Chen et al., 2024a). As shown in Table 17, Q&C consistently restores or improves performance across all metrics, even when the underlying INT4 quantizer and DeepCache introduce severe degradation.

| Method | MJHQ Metrics | | | |
|---|---|---|---|---|
| | **FID** ($\downarrow$) | **IR** ($\uparrow$) | **LPIPS** ($\downarrow$) | **PSNR** ($\uparrow$) |
| **SVDQuant (FLUX.1) (INT W4A4)** | 19.9 | 0.935 | 0.223 | 21.0 |
| **SVDQuant + DeepCache (FLUX.1)** | 23.85 | 0.890 | 0.277 | 18.9 |
| **SVDQuant + DeepCache + Q&C (FLUX.1)** | **20.5** | **0.93** | **0.23** | **20.6** |
| **SVDQuant (PixArt-Σ) (INT W4A4)** | 19.2 | 0.878 | 0.323 | 17.6 |
| **SVDQuant + DeepCache (PixArt-Σ)** | 25.6 | 0.762 | 0.585 | 14.3 |
| **SVDQuant + DeepCache + Q&C (PixArt-Σ)** | **20.1** | **0.87** | **0.323** | **17.9** |

Table 17: Evaluation on FLUX.1 and PixArt-

These results confirm that both the observed inconsistency problem and our proposed Q&C solution **generalize reliably across architectures, quantizers, and model families**.

## L IMAGE GENERATIONS WITH OUR METHOD ON DIT.

Figure 7 Supplementary visualization results of our method, accelerated by 12.7 $\times$ compared to the 250-step DiT model, on ImageNet 256×256. Our method generates outputs that closely resemble those of the 250-step model, maintaining high fidelity to the original results.

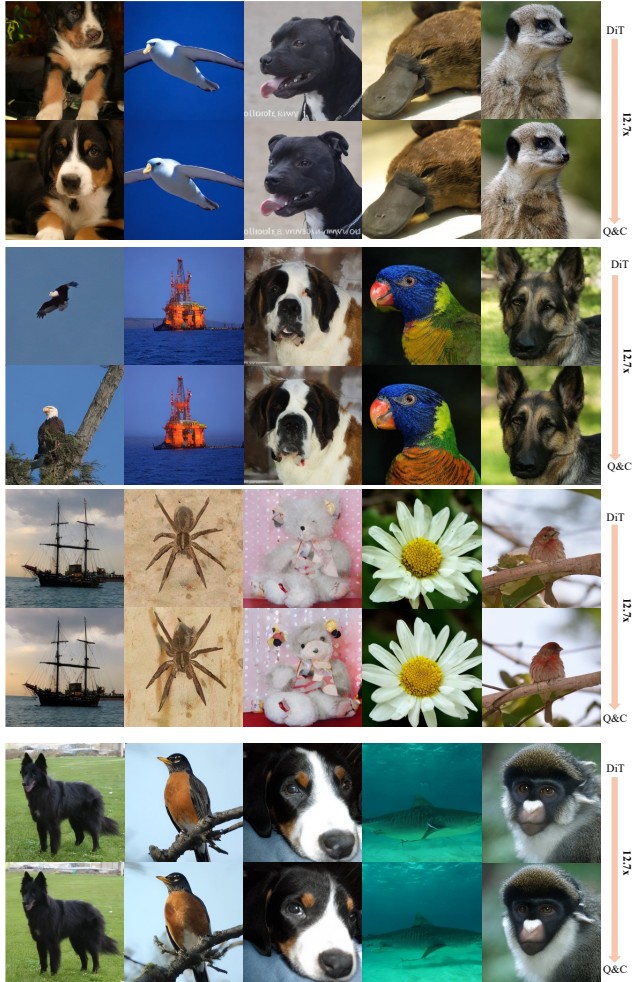

Figure 7: Image generations with our method on DiT. The image sizes are $256 \times 256$. The control group consists of images generated using DDPM with 250 steps.

