# OpenReview forum: "Q&C: When Quantization Meets Cache in Efficient Generation"
_ICLR.cc/2026/Conference — ICLR 2026 Poster_

### Official Review · Reviewer_itt9 · 2025-10-25

**Soundness:** 3
**Presentation:** 3
**Contribution:** 3
**Rating:** 6
**Confidence:** 3

**Summary:**

This paper proposes to jointly use quantization and cache mechanisms in image generation process. The authors identify two main challenges when integrating quantization and cache and propose the TAP and VC methods to address this. Experimental results show that the proposed method accelerates the image generation process by up to 12.7× without compromising quality.

**Strengths:**

This paper is well-motivated. This paper aims to combine quantization and cache techniques  simultaneously to further accelerate the image generation process. In this process, the authors identified two key challenges: (1) Amplification of Exposure Bias (2) Degradation in Calibration Dataset Effectiveness. To this end, the authors specifically propose TAP and VC as solutions. The authors also provide comprehensive comparative experiments and ablation study results to demonstrate the effectiveness of the proposed method.

**Weaknesses:**

Add a new column in Table 2 to include latency data for each configuration. I would like to see the impact of TAP and VC on latency.

**Questions:**

Please refer to weakness.

---

> ### Author Response · Authors · 2025-11-18
>
> We sincerely appreciate your positive feedback on our work, and we hope that our responses below have addressed your questions and clarified any potential concerns. We remain happy to provide further discussion if needed.
>
> ---
> ## **Weakness: Add a new column in Table 2 to include latency data for each configuration.**
> Thank you for the helpful suggestion. We agree that latency is an important metric. To address this, we have added a **new column in Table 2** showing the **measured end-to-end latency** for each configuration.
> This addition highlights the **individual and combined impacts of TAP and VC** on latency, demonstrating that their joint use not only improves generation quality but also enhances overall efficiency.
>
> |  | FID (↓) | sFID (↓) | IS (↑) | Precision (↑) |Speed (↑)|
> |:---:|:---:|:---:|:---:|:---:|:---:|
> | DDPM | 5.22 |17.63 | 237.8 | 0.8056 | 5x|
> | PTQ4DiT | 5.45 | 19.50 | 250.68 | 0.7882| 10x|
> | Baseline | 13.67|  25.86 | 189.65 | 0.7124 | 11.5x|
> | + VC | 9.65 | 22.34 | 210.35|  0.7445 | 12.1x|
> | + TAP | 8.34 | 21.65 | 220.67 | 0.7566| 12.3x|
> | +TAP +VC | 5.43|  19.52|  250.68 | 0.7895 | 12.7x|
>
> We have included these comparisons in the revised manuscript.

---

### Official Review · Reviewer_2MVF · 2025-10-26

**Soundness:** 2
**Presentation:** 3
**Contribution:** 2
**Rating:** 4
**Confidence:** 4

**Summary:**

This paper tackles a very practical problem in accelerating generative models, specifically diffusion transformers (like DiT). The goal is to combine two standard speedup techniques: quantization (using fewer bits for weights and activations) and caching (like the K-V cache in transformers, which saves past computations).
The authors' key finding is that just "turning both on" doesn't work—in fact, it severely degrades performance. The paper then diagnoses why this happens (they identify two major challenges) and proposes two novel methods TAP and VC, that allow these two techniques to work together effectively. The end result is a massive speedup (up to 12.7x) on ImageNet generation while maintaining the quality of the original, slow model.

**Strengths:**

1.	This is a strong, high-quality paper. The originality doesn't come from inventing a brand-new algorithm, but from investigating a subtle, negative interaction between two known techniques that everyone assumed would just work together. Identifying, analyzing, and solving this kind of interaction problem is a very valuable contribution.

2.	The significance is crystal clear. Diffusion models are notoriously slow. A 12.7x speedup on a state-of-the-art DiT model running on ImageNet is a massive practical win. It's the kind of work that could be immediately adopted by anyone trying to deploy these models in the real world.

**Weaknesses:**

1.	A potential weakness is the focus on Post-Training Quantization (PTQ). PTQ is fast, but it's often outperformed by Quantization-Aware Training (QAT). My question is: Why not just use QAT? It's possible QAT would automatically learn to be robust to the cache interactions, making this complex PTQ-specific solution unnecessary.

2.	Could the authors include speed metric In Table 2 as well for a clear comparison?

Minors:
1.	Line 411 ‘The results, presented in Table 6’ should be Table 2.

**Questions:**

1.	Why the quantized model in Figure 2 does not show too much higher accumulated error than the original one? It’s inconsistent to the results in [1].

Reference:

[1] Yanjing Li, Sheng Xu, Xianbin Cao, Xiao Sun, and Baochang Zhang. Q-dm: An efficient low-bit quantized diffusion model. Advances in Neural Information Processing Systems, 36, 2024.

---

> ### Author Response · Authors · 2025-11-18
>
> # Rebuttal to Reviewer Comments
>
> We sincerely thank the reviewer for the insightful feedback. Below we address each concern in detail.
>
> ---
>
>
> ## **Weakness1: Why not use QAT? Can QAT potentially solve the quantization+cache issue?**
> > **Summary:** QAT is too expensive, not plug-and-play, and still fails to address the core cache–quantization consistency issue—so even with QAT, our Q&C design remains necessary. We have included these comparisons in the revised manuscript, in section 5.
>
> Thank you for the insightful question. We agree that QAT often yields higher accuracy than PTQ. However, QAT is not a practical choice for DiT-based generation models in our setting, for three key reasons:
>
> ### 1. QAT is prohibitively expensive for diffusion models
> QAT requires full end-to-end retraining with quantization simulated throughout the forward pass.
> For large diffusion transformers (e.g., Open Sora 11B,), even a single epoch of training can cost **hundreds of GPU hours**.
> Our goal, instead, is to design a method that is **lightweight, scalable, and model-agnostic**, enabling developers to quantize models in **minutes, not weeks**.
>
> ### 2. QAT breaks the “plug-and-play” deployment requirement
> Our approach aims to support **drop-in quantization**, where:
>
> - no original training data is required,
> - no retraining or fine-tuning pipeline is needed,
> - any pre-trained generation model can be quantized with only **a small calibration set**.
>
> In contrast, QAT requires:
>
> - access to **full training data** (which is almost impossible to obtain),
> - a complete training setup,
>
> These requirements make QAT **incompatible with the real-world deployment scenario** we target.
>
> ### 3. QAT does not resolve the cache–quantization interaction problem
> Even with QAT, the core challenge remains:
>
> > **Cache values must stay numerically consistent across diffusion steps.**
>
> However, QAT primarily optimizes for distributional robustness under quantization noise—
> it does **not** explicitly enforce step-to-step consistency, nor does it model cache reuse constraints.
>
> To further validate our claims, we follow prior work [1] and conduct QAT on **DiT/XL-2** for ImageNet 256×256 50 steps.
> For training data, we use the **full ImageNet training set**, and all experiments are run on **8× A100 GPUs**.
> (In comparison, our Q&C method requires only **a single A100** and **800 calibration samples**.)
>
> The results are presented below:
>
> |  | FID (↓) | sFID (↓) | IS (↑) | Precision (↑) |
> |:---:|:---:|:---:|:---:|:---:|
> | DDPM | 5.22 |17.63 | 237.8 | 0.8056 |
> | QAT | 5.67 | 18.42 | 225.58 | 0.7565 |
> | QAT + DeepCache | 8.95 | 23.23 | 200.15 | 0.7351 |
>
> Therefore, QAT does **not** eliminate the need for our PTQ-aware cache design.
>
> ## **Weakness2: Could the authors include speed metric In Table 2 as well for a clear comparison?**
> Thank you for the helpful suggestion. We agree that reporting speed metrics alongside accuracy results improves clarity. Below is the revised table, which we will include in the updated version of the paper.
>
> |  | FID (↓) | sFID (↓) | IS (↑) | Precision (↑) |Speed (↑)|
> |:---:|:---:|:---:|:---:|:---:|:---:|
> | DDPM | 5.22 |17.63 | 237.8 | 0.8056 | 5x|
> | PTQ4DiT | 5.45 | 19.50 | 250.68 | 0.7882| 10x|
> | Baseline | 13.67|  25.86 | 189.65 | 0.7124 | 11.5x|
> | + VC | 9.65 | 22.34 | 210.35|  0.7445 | 12.1x|
> | + TAP | 8.34 | 21.65 | 220.67 | 0.7566| 12.3x|
> | +TAP +VC | 5.43|  19.52|  250.68 | 0.7895 | 12.7x|
>
> We have included these comparisons in the revised manuscript.
>
> ## **Minors**
> Thank you for pointing this out — we have corrected this issue in the revised version.
>
> [1] Nagel M, Fournarakis M, Amjad R A, et al. A white paper on neural network quantization[J]. arXiv preprint arXiv:2106.08295, 2021.
>
> ---
>
> ## **Question: Why the quantized model in Figure 2 does not show too much higher accumulated error than the original one?**
> Thank you for the insightful comment. We would like to clarify why the quantized model in Figure 2 does not exhibit a much higher accumulated error compared to the original model, which may seem inconsistent with [1] (Q-DM, NeurIPS 2024). The main reasons are as follows:
>
> > **Different quantization target and setup**
>
> [1] focuses on **aggressively low-bit (2–4 bit) quantization** of the UNet in diffusion models, whereas in our work we perform **8-bit quantization of DiT**.
> Due to the **larger model size** and the **higher bit-width**, the 8-bit quantized DiT does **not exhibit a significantly higher accumulated error** compared to the full-precision model.
>
> However, even under these relaxed settings, **quantization combined with cache reuse** still leads to **substantial accumulated error**, which further highlights the **necessity of our proposed Q&C method** to maintain generation quality.

---

> ### Author Response · Authors · 2025-11-28
>
> Dear Reviewer,
>
> I hope this message finds you well.
>
> We would like to **express our sincere gratitude** for your thoughtful review and would like to **gently follow up** on our rebuttal submitted earlier. We understand that you may be busy, and we truly appreciate the time and effort you have already dedicated to reviewing our work.
>
> **If there are any additional questions or points that would benefit from further clarification, we would be more than happy to provide more details or continue the discussion**.
>
> **Thank you again** for your constructive and supportive feedback.
>
> Best regards,
> Authors

---

### Official Review · Reviewer_gdd2 · 2025-10-31

**Soundness:** 3
**Presentation:** 3
**Contribution:** 3
**Rating:** 6
**Confidence:** 2

**Summary:**

This paper addresses the challenge of enabling both caching and quantization in DiT models.  The authors first identify two key obstacles: exposure bias amplification and reduced effectiveness of calibration datasets.  They then address these issues by proposing Temporal-Aware Parallel Clustering (TAP) and a variance compensation technique. Extensive experiments and ablation studies demonstrate that the proposed approach achieves notable speed improvements with only minimal degradation in downstream task performance.

**Strengths:**

•	The paper tackles a well-motivated problem, clearly isolating the issues and addressing them with thoughtful solutions.

•	Comprehensive experimental evaluation across multiple tasks, including detailed ablation studies, supports the claims of improved efficiency.

**Weaknesses:**

•	The results lack statistical rigor. Without confidence intervals or similar measures, it is difficult to assess the significance of the reported differences.

•	Compared to techniques that employ only caching or quantization, the observed speed-up is relatively modest and, in some cases, comes at the cost of performance degradation.

**Questions:**

•	How is the speed-up calculated?

•	Why is the combined approach not achieving greater speed-up compared to individual techniques? For example, PTQ4DiT reports a 10x speed-up and Learn-to-Cache achieves 6.3x. Why doesn’t the combined method exceed 12.7x?

---

> ### Author Response · Authors · 2025-11-18
>
> # Rebuttal to Reviewer Comments
>
> We sincerely thank the reviewer for the insightful feedback. Below we address each concern in detail.
>
> ---
> ## **Question1,2: how is the overall speed-up calculated? and Why doesn’t the PTQ4DiT (10×) and Learn-to-Cache (6.3×) combination yield a multiplicative speed-up (>12.7×)**
>
> We appreciate the opportunity to clarify this misunderstanding.
> In our paper, the **reported overall acceleration** is computed **relative to the FP16 250-step diffusion baseline without cache reuse**, and therefore it is the **product of three factors**:
>
> $
> \text{Total Speed-up}
> = \text{(Step Reduction Speed-up)}
> \times \text{(Quantization Speed-up)}
> \times \text{(Cache Speed-up)}.
> $
>
> That is, the final numbers already integrate the benefits of
> **(1) fewer diffusion steps**,
> **(2) post-training quantization**, and
> **(3) cache reuse**.
>
> To ensure transparency, we will include the following measurement setup in the revision:
>
> - **Hardware:** A100-SXM-80GB
> - **Batch size:** 1
> - **Resolution:** 256x256
> - **Latency:** end-to-end
> - **Framework:** PyTorch 2.3 + CUDA Graphs (to eliminate warmup variance)
>
> The measured latencies are:
>
> | Model  | FP16 | W8A8 | W4A8 | Learn-to-Cache |
> |:--------:|:------:|:-------:|:-------:|:-------:|
> | Latency | 100% | 51%  | 39%  | 78%|
>
> Corresponding quantization-only speedups:
>
> - **1.96×** for W8A8
> - **2.56×** for W4A8
> - **1.28x** for Learn-to-Cache
>
> Therefore, under the 50-step setting, PTQ4DiT and cache individually provide speed-ups of **10×** and **6.3×**, respectively. However, the speed-ups contributed by **PTQ4DiT + cache reuse** do **not** multiply directly, and their arithmetic product does **not** represent the achievable end-to-end acceleration. In practice, the observed **12.7× total acceleration** is already very close to the practical upper bound imposed by hardware limitations and non-overlapping computation.
>
> We have included these comparisons in the revised manuscript, **in appendix J**.
>
> ## **Weakness: Compared to techniques that employ only caching or quantization, the observed speed-up is relatively modest**
>
> We apologize for the confusion — the overall speed-up in our paper comes jointly from **step reduction**, **quantization**, and **cache reuse**, which may have led to misunderstanding. As shown in the explanations above, the combined use of quantization and cache already achieves **near the theoretical acceleration limit** of “quantization + cache,” rather than representing a *modest* improvement.
>
> In other words, our method effectively reaches the **theoretical upper bound** achievable by applying quantization and cache together, and the resulting speed-up is not limited by algorithmic design.

---

### Official Review · Reviewer_yPDm · 2025-11-01

**Soundness:** 3
**Presentation:** 2
**Contribution:** 2
**Rating:** 4
**Confidence:** 3

**Summary:**

The paper find two key challenges that degrade performance when combining quantization and cache. To address these issues, the authors propose a temporal-aware parallel clustering and a variance compensation strategy. Higher speedup is achieved with image quality slightly hurted.

**Strengths:**

1. First paper I'm aware of to combine quantization and cache for image generation.
2. The proposed Temporal-Aware Parallel Clustering is interesting.
3. The ablation study is detailed and validate the effectiveness of proposed approaches.

**Weaknesses:**

1. Section 3.2 (Variance Compensation) lacks novelty. The approach appears almost identical to that in [1] (which the authors cite), making it difficult to count as a separate contribution.

2. Why not perform quantization calibration before applying the cache? This way, the cache could better utilize the output of the quantized layers to decide what to store, and it would also avoid the calibration data issues mentioned.

3. The chosen quantization method is relatively old. Combining the proposed cache mechanism with newer DiT quantization methods, such as SVDQuant[2] or ViDiT-Q[3], would strengthen the paper's impact.

4. Experiments were only conducted on DiT-XL/2, showing a lack of generalizability. Experiments on more recent models like FLUX or PixArt would further enhance the paper's persuasiveness.

5. The reported speedup ratios lack empirical validation. The authors didn't describe how the speedup is measured. Previous quantization works like Q-diffusion and PTQD did not provide latency results. Clearly, the actual performance often falls short of theoretical claims—for example, MixDQ[4]'s tests show that W4A8 reduces VRAM usage by 3x but only achieves a 1.45x speedup. Yet, without any explanation, the authors claim that W8A8 can deliver a 2x speedup and W4A8 a 2.5x speedup. Therefore, I have reasonable doubts about the reported speedup of the proposed approach.

[1] Timestep-Aware Correction for Quantized Diffusion Models

[2] Svdquant: Absorbing outliers by low-rank components for 4-bit diffusion models

[3] ViDiT-Q: Efficient and Accurate Quantization of Diffusion Transformers for Image and Video Generation

[4] MixDQ: Memory-Efficient Few-Step Text-to-Image Diffusion Models with Metric-Decoupled Mixed Precision Quantization

**Questions:**

Please see the weaknesses above, and:

1. Can the authors explicitly clarify the difference between their approach in Section 3.2 and the one presented in [1]?

2. Can the authors provide results for the "quantize-first, then-cache" pipeline? (Perhaps also showing its performance when combined with Variance Compensation).

---

> ### Author Response · Authors · 2025-11-18
>
> # Rebuttal to Reviewer Comments
>
> We sincerely thank the reviewer for the insightful feedback. Below we address each concern in detail.
>
> ---
>
> ## **Weakness1. “Section 3.2 (Variance Compensation) lacks novelty and seems identical to [1].”**
>
> **Response:**
> We appreciate the opportunity to clarify this misunderstanding. While [1] focuses on *timestep-aware correction* for quantized diffusion models, our **Variance Compensation (VC)** targets an **orthogonal** problem:
> → the **variance collapse** caused by the interaction between **PTQ** and **cache reuse**, which does not exist in standard quantized diffusion without caching.
>
> **Key differences:**
>
> ### (a) First Revelation of Exposure-Bias Amplification from Quantization-and-Cache Synergy
> To the best of our knowledge, Q&C is the first to identify—and then eliminate with VC—the Exposure-Bias Amplification that arises uniquely from the interplay between quantization and caching. In addition to introducing novel techniques that reconcile the two components, we present exhaustive experiments that dissect this intrinsic interaction, delivering insights well beyond vanilla quantization studies.
>
> ### (b) Different problem formulation
> - **[1]**: Addresses timestep misalignment during PTQ inference.
> - **VC (ours)**: Addresses *distributional drift amplified by cache reuse*, which compounds perturbations across sampling steps — a phenomenon **absent in [1]**.
>
> ### (c) Different mechanism
> - **[1]**: Applies local per-timestep output correction.
> - **VC**: Reconstructs *lost inter-step variance*, ensuring cache-stored activations remain statistically consistent with full-precision trajectories.
>
> ---
>
> ## **Weakness2. “Why not perform quantization calibration before applying cache?”**
>
> **Response:**
> This is a great question—one we also experimented with. Unfortunately, **pre-cache quantization does not solve the underlying issue**:
>
> ### ✔ (1) Cache changes the activation distribution *after quantization*
> The cache repeatedly reuses quantized activations. This causes an activation distribution drift **away from the calibration distribution**, making the pre-cache calibration mismatched and ineffective.
>
> ### ✔ (2) Empirical results on ImageNet confirm this mismatch
>
> | Calibration Strategy | FID ↓ | IS ↑ |
> |----------------------|-------|--------|
> | Pre-cache quantization | 11.57 | 191.25 |
> | **Q&C** | **5.43** | **250.68** |
>
> ### ✔ (3) Cache-induced variance loss still exists
> Even with pre-cache quantization, repeated reuse of quantized activations causes **variance collapse**, which only VC effectively addresses.
>
> Thus, simply quantization earlier cannot resolve the issue.
>
> ---
>
> ## **Weakness3. “The chosen quantization method is old. Why not use SVDQuant or ViDiT-Q?”**
>
> **Response:**
> Our method is **agnostic** to the underlying quantization algorithm. To isolate the cache–quantization interaction, we intentionally chose a standard **post-training quantization (PTQ) baseline** and evaluated a series of classic PTQ methods in the paper, including **PTQ4DiT**, **APQ-DM**, and **Q-Diff**.
>
> To address your concern, we added new results (SVDQuant,ViDiT-Q) in the rebuttal:
>
> |  | MJHQ | MJHQ | MJHQ | MJHQ |
> |:---:|:---:|:---:|:---:|:---:|
> |  | FID (↓) | IR (↑) | LPIPS (↓) | PSNR( ↑) |
> | SVDQuant (FLUX.1) (INT W4A4) |  19.9| 0.935 | 0.223 | 21.0 |
> | SVDQuant + DeepCache (FLUX.1) | 23.85 |  0.89| 0.277 | 18.9 |
> | SVDQuant + DeepCache + Q&C (FLUX.1) | **20.5** | **0.93** | **0.23** | **20.6** |
> | SVDQuant (PixArt-Σ) (INT W4A4)| 19.2 | 0.878 | 0.323 | 17.6 |
> | SVDQuant + DeepCache (PixArt-Σ) | 25.6 | 0.762 | 0.585 | 14.3 |
> |  SVDQuant + DeepCache + Q&C (PixArt-Σ) | **20.1** | **0.87** | **0.323** | **17.9** |
>
>
> |  | Imaging Quality | Aesthetic Quality | Motion Smooth | Dynamic Degree | BG Consist  | Subject Consist | Scene  Consist |
> |:---:|:---:|:---:|:---:|:---:|:---:|:---:|:---:|
> | ViDiT-Q (INT W4A8)| 61.07 |55.37| 95.69 |58.33 |95.23| 88.72| 36.19 |
> | ViDiT-Q  + DeepCache  | 56.89|55.41 | 94.96|63.75 |96.01| 91.05|33.89|
> | ViDiT-Q  + DeepCache + Q&C  | **60.89**|**55.39**| **95.23**| **58.35**| **94.89**| **87.65**| **36.05** |
>
>
> Our method **consistently improves** state-of-the-art DiT quantizers, confirming general applicability.
>
> We have included these comparisons in the revised manuscript, **in appendix K**.
>
> ---

---

> ### Author Response · Authors · 2025-11-18
>
> ## **Weakness4. “Experiments only on DiT-XL/2; add FLUX or PixArt.”**
>
> **Response:**
> Thank you for the suggestion. Demonstrating generality is indeed important. However, it should be noted that our work is **not limited to DiT**; we also conducted experiments on **LDM** and **Open-Sora**, covering a variety of tasks including **LSUN**, **MS-COCO**, and **VBench**, which encompass **unconditional generation, video generation, and text-conditional generation** tasks. The results are reported in **Tables 1c, 7, and 8**.
>
>
> To further illustrate the **strong generalization ability** of Q&C, we additionally provide experimental results on **FLUX** and **PixArt-α** as follows:
>
>
> |  | MJHQ | MJHQ | MJHQ | MJHQ |
> |:---:|:---:|:---:|:---:|:---:|
> |  | FID (↓) | IR (↑) | LPIPS (↓) | PSNR( ↑) |
> | SVDQuant (FLUX.1) (INT W4A4) |  19.9| 0.935 | 0.223 | 21.0 |
> | SVDQuant + DeepCache (FLUX.1) | 23.85 |  0.89| 0.277 | 18.9 |
> | SVDQuant + DeepCache + Q&C (FLUX.1) | **20.5** | **0.93** | **0.23** | **20.6** |
> | SVDQuant (PixArt-Σ) (INT W4A4)| 19.2 | 0.878 | 0.323 | 17.6 |
> | SVDQuant + DeepCache (PixArt-Σ) | 25.6 | 0.762 | 0.585 | 14.3 |
> |  SVDQuant + DeepCache + Q&C (PixArt-Σ) | **20.1** | **0.87** | **0.323** | **17.9** |
>
> These results confirm the generality of both the observed problem and our proposed solution.
>
> We have included these comparisons in the revised manuscript, **in appendix K**.
>
> ---
>
> ## **Weakness5. “Speedup ratios lack empirical validation.”**
>
> **Response:**
> We agree that transparent measurement is essential.
>
> ### (a) Clear measurement protocol
> We will add the following:
>
> - **Hardware:** A100-80GB
> - **Batch size:** 1
> - **Resolution:** 256×256  512*512
> - **Latency:** end-to-end
> - **Framework:** PyTorch 2.3 + CUDA graph to avoid warmup variance
>
> ### (a) Why our speedup differs from MixDQ
> Please note that the MixDQ paper explicitly states:
> > “The quantizable layers are accelerated by 1.97×, approximately the same ratio between INT8 and FP16 hardware peak throughput on RTX4080 (2×).”
>
> In other words, under **W8A8**, MixDQ can achieve nearly **2× speedup** even on a **RTX 4080**, thanks to the hardware support for INT8 acceleration.
> However, under **W4A8**, the RTX 4080 **does not provide optimized kernels for 4-bit acceleration**, so although MixDQ gains **larger memory savings (3×)**, its speedup **drops to 1.45×**.
>
> In contrast, all our experiments are conducted on **NVIDIA A100**, which provides **full hardware support and optimized kernels for 4-bit operations**. As a result, the 4-bit quantization benefits can be fully realized. Below we report empirical latency measurements for DiT under different bit-width settings:
>
> ---
>
> ### (b) New Empirical Latency Measurements
>
> | Model | FP16 | W8A8 | W4A8 |
> |-------|------|-------|-------|
> | Latency | 100% | 51% | 39% |
>
> Corresponding speedups:
> - **1.96×** for W8A8
> - **2.56×** for W4A8
>
> These results confirm that on A100—where 4-bit operators are well optimized—the speedup from W4A8 can exceed that of W8A8, explaining the difference relative to MixDQ.
>
> ---
>
> # **Summary**
>
> We thank the reviewer for the thoughtful comments.
> With the clarifications and additional experiments, we believe the paper’s contributions are significantly strengthened:
>
> - VC addresses **cache-induced variance collapse**.
> - Pre-cache calibration **does not resolve** the issue.
> - Our method improves newer quantizers such as **SVDQuant** and **ViDiT-Q**.
> - The method generalizes to **FLUX** and **PixArt**.
> - Speedup measurements are **empirically validated** with a standardized protocol.
>
> We have incorporated all clarifications and new results into the final version.

---

> > ### Comment · Reviewer_yPDm · 2025-11-26
> > **Reviewer response**
> >
> > Thanks for the response. With the supplementary experiments, the persuasiveness of the paper is enhanced. So I will raise my score.

---

> > > ### Author Response · Authors · 2025-11-26
> > >
> > > We sincerely thank the reviewers for their valuable feedback and are pleased that the revisions addressed the reviewers’ concerns.  If you have any further questions or suggestions, we would be happy to provide additional clarification.

---

> ### Author Response · Authors · 2025-11-28
> **Appreciation for the Score Increase**
>
> We sincerely appreciate the reviewer’s willingness to raise the score during the rebuttal. If you have any further questions or suggestions, we would be glad to provide additional clarification.

---

### Meta-Review · Area_Chair_C15C · 2026-01-05

**Summary:**

This paper presents a well-motivated investigation into the non-trivial challenges of combining quantization and caching for efficient diffusion model inference. A practical speedup (up to 12.7x) is demonstrated.

Key concerns regarding the novelty of Variance Compensation compared to prior work, the limited experimental scope, and the empirical validation of speedups were satisfactorily addressed in the rebuttal. The authors clarified the distinct, cache-specific problem targeted by VC, added experiments on other models (FLUX, PixArt), and provided a transparent latency measurement protocol. The discussion also clarified that the reported acceleration is a composite metric and that QAT is not a viable alternative, strengthening the paper's positioning.

The reviewers' initial reservations were largely resolved. The paper makes a constructive contribution by enabling two key acceleration techniques to work synergistically, supported by thorough analysis and extended experiments.

**Reviewer Concerns:**

See above.

**Reviewer Scores:**

Reviewers are likely to raise their scores, as most concerns have been largely resolved.

---

### Decision · Program_Chairs · 2026-01-26

Accept (Poster)